# ORIENT ANYTHING

## ABSTRACT

Orientation estimation is a fundamental task in 3D shape analysis which consists of estimating a shape's orientation axes: its side-, up-, and front-axes. Using this data, one can rotate a shape into canonical orientation, where its orientation axes are aligned with the coordinate axes. Developing an orientation algorithm that reliably estimates complete orientations of general shapes remains an open problem. We introduce a two-stage orientation pipeline that achieves state of the art performance on up-axis estimation and further demonstrate its efficacy on full-orientation estimation, where one seeks all three orientation axes. Unlike previous work, we train and evaluate our method on all of Shapenet rather than a subset of classes. We motivate our engineering contributions by theory describing fundamental obstacles to orientation estimation for rotationally-symmetric shapes, and show how our method avoids these obstacles.

## 1 INTRODUCTION

*Orientation estimation* is a fundamental task in 3D shape analysis which consists of estimating a shape's orientation axes: its side-, up-, and front-axes. Using this data, one can rotate a shape into *canonical orientation*, in which the shape's orientation axes are aligned with the coordinate axes. This task is especially important as a pre-processing step in 3D deep learning, where deep networks are typically trained on datasets of canonically-oriented shapes but applied to arbitrarily-oriented shapes at inference time. While data augmentation or equivariant and invariant architectures may improve a model's robustness to input rotations, these techniques come at the cost of data efficiency and model expressivity (Kuchnik & Smith, 2019; Kim et al., 2023). In contrast, orientation estimation allows one to pre-process shapes at inference time so that their orientation matches a model's training data.

Orientation estimation is a challenging task, and developing an orientation pipeline that reliably estimates complete orientations of general shapes remains an open problem. The naïve deep learning approach is to train a model with an $L^2$ loss to directly predict a shape's orientation from a point cloud of surface samples. However, this strategy fails for shapes with rotational symmetries, where the optimal solution to the $L^2$ regression problem is the *Euclidean mean* (Moakher, 2002) of a shape's orientations over all of its symmetries. In contrast, works such as Poursaeed et al. (2020) discretize the unit sphere into a set of fixed rotations and train a classifier to predict a probability distribution over these rotations, but find that this approach fails for any sufficiently dense discretization of the unit sphere.

Our key insight is to divide orientation estimation into two tractable sub-problems. In the first stage (the *quotient orienter*), we solve a continuous regression problem to recover a shape's orientation *up to octahedral symmetries*. In the second stage (the *flipper*), we solve a discrete classification problem to predict one of 24 octahedral flips that returns the first-stage output to canonical orientation. Octahedral symmetries form a small set covering a substantial proportion of the symmetries occurring in real-world shapes. Consequently, quotienting our first-stage regression problem by octahedral symmetries prevents its predictions from collapsing to averages, while also keeping the subsequent classification problem tractable.

Using this strategy, our method achieves state-of-the-art performance on the well-studied problem of up-axis prediction, and additionally performs well on full-orientation prediction, which few prior works have tackled. Unlike previous work, we train and evaluate our model on the *entire* Shapenet dataset rather than a subset of classes. We further demonstrate its generalization capabilities on Objaverse, a large dataset of real-world 3D models of varying quality.

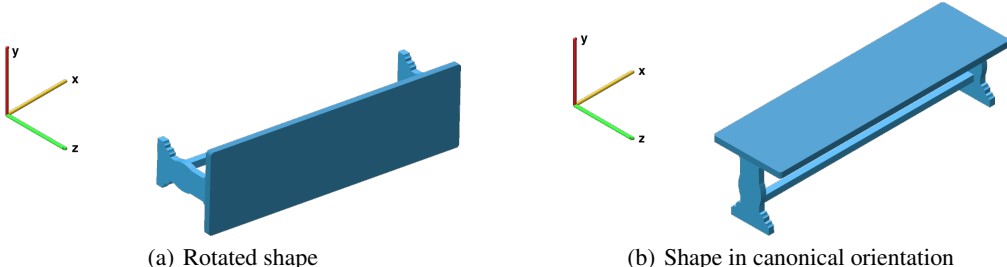

(a) Rotated shape          (b) Shape in canonical orientation

Figure 1: *Orientation estimation* allows users to rotate arbitrary shapes (a) into canonical orientation (b), in which the shape's orientation axes are aligned with the coordinate axes.

A shape's ground truth orientation may be ambiguous. This challenge is especially salient for nearly-symmetric shapes, where multiple orientations may yield nearly indistinguishable shapes. To resolve this issue, we use conformal prediction to enable our flipper to output *adaptive prediction sets* (Romano et al., 2020) whose size varies with the flipper model's uncertainty. For applications with a human in the loop, this enables the end user to choose from a small set of plausible candidate orientations, dramatically simplifying the orientation estimation task while preserving user control over the outputs.

Our contributions include the following: (1) we identify fundamental obstacles to orientation estimation and study the conditions under which a naïve regression-based approach to orientation estimation fails; (2) we propose a two-stage orientation estimation pipeline that sidesteps these obstacles; (3) we train and test our model on Shapenet and show that it achieves SOTA performance for orientation estimation; (4) we use conformal prediction to enable end users to resolve ambiguities in a shape's orientation; (5) we release our code and model weights to share our work with the ML community.

## 2 RELATED WORK

**Classical methods.** A simple method for orientation estimation is to compute a rotation that aligns a shape's principal axes with the coordinate axes; Kaye & Ivrissimtzis (2015) propose a robust variant of this method for mesh alignment. However, Kazhdan et al. (2003) find that PCA-based orientation estimation is not robust to asymmetries. Jin et al. (2012); Wang et al. (2014) propose unsupervised methods that leverage low-rank priors on axis-aligned 2D projections and third-order tensors, respectively, constructed from input shapes. These priors are restrictive, and the resulting orientation pipelines also fail on asymmetric shapes.

Another set of classical methods observe that as many man-made objects are designed to stand on flat surfaces, their up axis is normal to a *supporting base*. Motivated by this observation, these methods attempt to identify a shape's supporting base rather than directly infer their up axis. Fu et al. (2008) generate a set of candidate bases, extract geometric features, and combine a random forest and SVM to predict a natural base from the candidates. Lin & Tai (2012) simplify a shape's convex hull, cluster the resulting facets to obtain a set of candidate bases, and compute a hand-designed score to select the best base. Both of these methods rely heavily on feature engineering and fail on shapes that do not have natural supporting bases.

**Deep learning-based methods.** Motivated by the limitations of classical approaches, several works use deep learning for orientation estimation. Liu et al. (2016) train two neural networks on voxel representations of 3D shapes. A first-stage network assigns each shape to one of $C$ classes. Based on this prediction, the shape is routed to one of $C$ second-stage networks that are independently trained to predict the up axis from voxel representations of shapes in their respective classes. This method is unable to handle shapes that lie outside the $C$ classes on which the networks were trained.

Pang et al. (2022) draw inspiration from classical methods and train a segmentation network to predict points that belong to a shape's supporting base. They fit a plane to the predicted base points

and output a normal vector to this plane as the predicted up axis. This method represents the current state of the art for orientation estimation, but struggles to handle shapes without well-defined natural bases and and only predicts a shape's up axis. In contrast, our method succeeds on general shapes and predicts a full rotation matrix that returns a shape to canonical orientation.

Chen et al. (2021) use reinforcement learning to train a model to gradually rotate a shape into upright orientation. While this algorithm performs well, it is evaluated on few classes and is costly to train. Kim et al. (2020) adopt a similar perspective to Fu et al. (2008), but use ConvNets to extract features for a random forest classifier that predicts a natural base. Poursaeed et al. (2020) use orientation estimation as a pretext task to learn features for shape classification and keypoint prediction. They also investigate a pure classification-based approach to orientation estimation that discretizes the 3D rotation group into $K$ rotations and predicts a distribution over these rotations for an arbitrarily-rotated input shape. They find that its performance decays rapidly as $K$ increases, reaching an accuracy as low as 1.6% for $K = 100$ rotations.

We also highlight a related literature on *canonical alignment*. This literature includes works such as Kim et al. (2023); Sajnani et al. (2022); Spezialetti et al. (2020); Zhou et al. (2022), which seek to map arbitrarily-rotated shapes to a class-consistent pose, as well as Katzir et al. (2022); Sun et al. (2021), which seek to learn pose-invariant representations of 3D shapes. These works only attempt to learn a consistent orientation within each class, but this orientation is not consistent across classes and is not generally aligned with the coordinate axes. In contrast, we tackle the more challenging task of inferring a canonical orientation that is consistent across *all* objects.

## 3 METHOD

In this section, we motivate and describe our orientation pipeline. We first identify fundamental obstacles to orientation estimation and show that learning a shape's orientation with the $L^2$ loss fails when the shape is rotationally symmetric. Motivated by these observations, we introduce our two-stage orientation pipeline consisting of a *quotient orienter* followed by a *flipper*. Our quotient orienter model solves a regression problem to recover a shape's orientation up to octahedral symmetries, which commonly occur in real-world shapes. The flipper then predicts one of 24 octahedral flips that returns the first-stage output to canonical orientation. We finally use conformal prediction to enable our flipper to output prediction sets whose size varies with the model's uncertainty. This allows end users to resolve ambiguities in a shape's orientation by choosing from a small set of plausible candidate orientations.

### 3.1 ORIENTATION ESTIMATION UNDER ROTATIONAL SYMMETRIES

In this section, we introduce the orientation estimation problem and motivate our approach. Throughout these preliminaries, we consider 3D shapes $S \in \mathcal{S}$ lying in some space of arbitrary shape representations $\mathcal{S}$. *Orientation estimation* consists of learning an *orienter* function $f : \mathcal{S} \to SO(3)$ that maps a shape $S \in \mathcal{S}$ to a predicted orientation $\hat{\Omega}_S \in SO(3)$, where $SO(3)$ denotes the 3D rotation group. An *orientation* is a rotation matrix $\Omega_S$ associated with a shape $S$ that is *rotation-equivariant*: If one rotates $S$ by $R \in SO(3)$ to obtain $RS$, then $\Omega_{RS} = R\Omega_S$. We interpret the columns of $\Omega_S = (\omega_S^x, \omega_S^y, \omega_S^z)$ as the side-, up-, and front-axes of $S$, respectively, and say that $S$ is in *canonical orientation* if $\Omega_S = I$. If $S$ is in canonical orientation, then its side-, up-, and front-axes are aligned with the $\{x, y, z\}$ coordinate axes, respectively. We depict a canonically-oriented shape $S$ along with its orientation $\Omega_S$ in Figure 2.

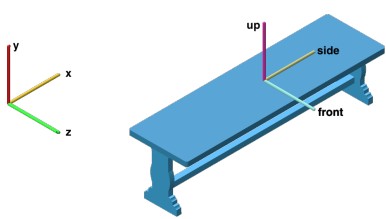

Figure 2: A shape's *orientation* $\Omega_S$ is a rotation matrix whose columns are the shape's side-, up-, and front-axes (plotted in yellow, magenta, cyan, resp).

Given a training set $\mathcal{D}$ of shapes $S \in \mathcal{S}$ paired with their ground truth orientations $\Omega_S$, a natural strategy for orientation estimation is to define a loss function $\ell$ on the space of orientations, parametrize $f$ as a neural network $f_\theta$, and solve the following problem:

$$\min_{\substack{f_\theta \\ R \sim U(SO(3)) \\ (S, \Omega_S) \in \mathcal{D}}} \mathbb{E} \left[ \ell \left( f_\theta(RS), R, \Omega_S \right) \right], \tag{1}$$

where $U(SO(3))$ is the uniform distribution over $SO(3)$. In the following section, we will motivate our approach by describing a theoretical obstacle to learning an orienter function $f$ *regardless* of one's choice of loss $\ell$, and then describe specific challenges associated with solving Equation 1 using the $L^2$ loss. These results will contextualize Propositions 3.3 and 3.4, which provide theoretical support for our two-stage orientation pipeline.

**Theoretical challenges.** An ideal orienter $f$ should satisfy two desiderata: (1) It should accept arbitrarily-oriented shapes $RS$ as input and output their orientation $\Omega_{RS} = R\Omega_S$, and (2) it should succeed on most shapes $S$ that occur in the wild. As many real-world shapes possess at least one non-trivial rotational symmetry, an ideal orienter should therefore succeed on rotationally-symmetric shapes. We begin by showing that no function can simultaneously satisfy these desiderata.

**Proposition 3.1** *Let $S \in \mathcal{S}$ be a fixed shape which is symmetric under a non-trivial group of rotations $\mathcal{R}_S \subseteq SO(3)$, and let $\Omega_S$ be its orientation. Then there is no function $f$ such that $f(RS) = R\Omega_S$ for all $R \in SO(3)$.*

We prove this result in Appendix A.1. Intuitively, if a shape $S$ is symmetric under some non-trivial rotation $R$, then $S$ is *invariant* under $R$, but its orientation is *equivariant* under $R$, so the map $RS \mapsto R\Omega_S$ is one-to-many and cannot be a function. This shows that any solution to the orientation estimation problem 1 necessarily trades off some desirable property; there are no functions that can successfully orient a rotationally-symmetric shape given any input orientation.

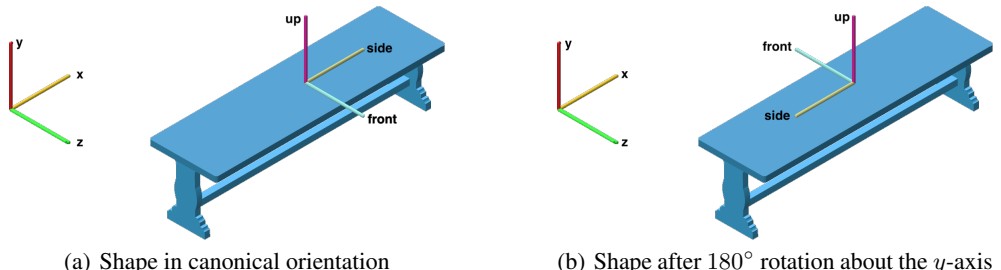

(a) Shape in canonical orientation  (b) Shape after $180°$ rotation about the $y$-axis

Figure 3: Rotating a shape by one of its symmetries changes its orientation while leaving the shape unchanged. Here, the front axis (in cyan) and side axis (in yellow) are flipped when the shape is rotated $180°$ about the $y$-axis.

Previous works such as Liu et al. (2016); Poursaeed et al. (2020) observe that orientation estimation via $L^2$ regression typically yields poor results. Motivated by this observation, we now characterize the solution to orientation estimation via $L^2$ regression for a single rotationally-symmetric shape and show that naïve $L^2$ regression degenerates in this setting.

**Proposition 3.2** *Let $S \in \mathcal{S}$ be a fixed shape which is symmetric under a non-trivial group of rotations $\mathcal{R}_S \subseteq SO(3)$. Let $\Omega_S$ be the shape's orientation, and suppose $f^* : \mathcal{S} \to SO(3)$ solves the following regression problem:*

$$\min_{f: \mathcal{S} \to SO(3)} \mathbb{E}_{R \sim U(SO(3))} \left[ \| f(RS) - R\Omega_S \|_F^2 \right], \tag{2}$$

*Then $f^*(RS) = proj_{SO(3)} \left[ \frac{1}{|\mathcal{R}_S|} \sum_{Q \in \mathcal{R}_S} RQ\Omega_S \right] \neq R\Omega_S$, where $proj_{SO(3)}$ denotes the orthogonal projection onto $SO(3)$.*

We prove this proposition in Appendix A.2. Proposition 3.2 shows that even when seeking to predict the rotated orientations $R\Omega_S$ of a *single* rotationally-symmetric shape $S$, $L^2$ regression fails to learn

the correct solution, and instead learns the *Euclidean mean* of the rotated orientations $RQ\Omega_S$ across all rotations $Q$ in the symmetry group $\mathcal{R}_S$ (Moakher, 2002).

This problem may be highly degenerate, even for shapes with a *single* non-trivial symmetry. For example, consider the bench shape $S$ depicted in Figures 1, 2, 3. As shown in Figure 3, this shape has two rotational symmetries: The identity rotation, and a $180°$ rotation about the $y$-axis. One may represent these rotations by the matrices $I$ and $Q := (-e_x, e_y, -e_z)$, resp., where $e_x, e_y, e_z$ are the standard basis vectors.

Proposition 3.2 states that one solves the $L^2$ regression problem 2 for the bench shape by computing the arithmetic mean of $I$ and $Q$ and then orthogonally projecting this matrix onto $SO(3)$. The arithmetic mean of $I, Q$ is the matrix $M := (0, e_y, 0)$, and one computes its orthogonal projection onto $SO(3)$ by solving a special Procrustes problem (Gower & Dijksterhuis, 2004). However, the solution to this problem is non-unique, and $f^*(S)$ may be any rotation about the $y$-axis, which we illustrate in Figure 4. This shows that even a single non-trivial rotational symmetry leads to *an entire submanifold of solutions* $f^*(S)$ to Problem 2.

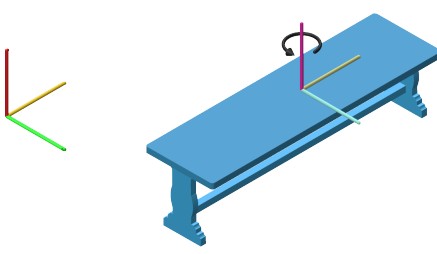

Figure 4: The solution $f^*(S)$ to Problem 2 evaluated at the bench shape $S$ may be any rotation about the $y$-axis.

**A partial solution.** The previous section shows that solving Equation 1 with the $L^2$ loss fails for rotationally-symmetric shapes, which are common in practice. We now present a partial solution to this problem. Suppose we know a finite group $\hat{\mathcal{R}} \supseteq \mathcal{R}_S$ that contains a shape $S$'s rotational symmetries. We can then *quotient* the $L^2$ loss by $\hat{\mathcal{R}}$ to obtain the following problem:

$$\min_{\substack{f:\mathcal{S}\to SO(3)}} \mathbb{E}_{\substack{R\sim U(SO(3)) \\ (S,\Omega_S)\in\mathcal{D}}} \left[\min_{Q\in\hat{\mathcal{R}}} \|f(RS) - RQ\Omega_S\|_F^2\right]. \qquad (3)$$

This loss is small if $f(RS)$ is close to the orientation $RQ\Omega_S$ of the rotated shape $RQS$ for *any* $Q \in \hat{\mathcal{R}}$; Mehr et al. (2018) use similar techniques to learn latent shape representations that are invariant under a group of geometric transformations. Intuitively, whereas Equation 2 attempts to make $f(S)$ close to *all* $Q\Omega_S$, a minimizer of Equation 3 merely needs to make $f(S)$ close to *any* $Q\Omega_S$. Formally:

**Proposition 3.3** *Let $S \in \mathcal{S}$ be a fixed shape which is symmetric under a group of rotations $\mathcal{R}_S \subseteq \hat{\mathcal{R}} \subseteq SO(3)$. Let $\Omega_S$ be the shape's orientation, and suppose $f^* : \mathcal{S} \to SO(3)$ is a solution to the following quotient regression problem:*

$$\min_{\substack{f:\mathcal{S}\to SO(3)}} \mathbb{E}_{R\sim U(SO(3))} \left[\min_{Q\in\hat{\mathcal{R}}} \|f(RS) - RQ\Omega_S\|_F^2\right], \qquad (4)$$

*Then for any $R \in SO(3)$, $f^*(RS) = RQ^*\Omega_S$ for some $Q^* \in \hat{\mathcal{R}}$.*

We prove this proposition in Appendix A.3. In contrast to naïve $L^2$ regression, quotient regression learns a function that correctly orients rotationally-symmetric shapes *up to a rotation* in the group $\hat{\mathcal{R}}$. While this is only a partial solution to the orientation estimation problem, the remainder reduces to a discrete classification problem: Predicting the rotation $Q^* \in \hat{\mathcal{R}}$ such that $f^*(RS) = RQ^*\Omega_S$. In the following section, we will show how a solution to this problem allows one to map $RS$ to the canonically-oriented shape $S$.

**Recovering an orientation via classification.** By solving the quotient regression problem in Equation 3, one can recover an arbitrarily-rotated shape $RS$'s orientation up to a rotation $Q^* \in \hat{\mathcal{R}}$. In this section, we propose training a classifier to predict this rotation $Q^*$ given the solution $f^*(RS) = RQ^*\Omega_S$ to the quotient regression problem. We now further assume that the shape

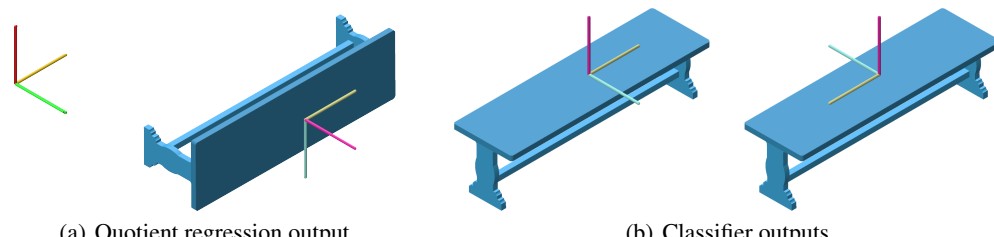

(a) Quotient regression output             (b) Classifier outputs

Figure 5: The quotient regression problem 4 correctly orients an arbitrarily rotated shape $RS$ up to a rotation in $\hat{\mathcal{R}}$. The classification problem 6 then recovers the orientation of $RS$ up to one of its rotational symmetries, which suffices for mapping $RS$ to the canonically-oriented shape $S$.

$S$'s ground truth orientation $\Omega_S$ is the canonical orientation $\Omega_S = I$. We show that even if $S$ is symmetric under some group of symmetries $\mathcal{R}_S \subseteq \hat{\mathcal{R}}$, the optimal classifier's predictions enable one to map $RS$ to the canonically-oriented shape $S$.

Predicting a rotation $Q^* \in \hat{\mathcal{R}}$ from the output $f^*(RS) = RQ^*$ of the quotient regression model is an $|\hat{\mathcal{R}}|$-class classification problem. One may train an appropriate classifier by solving the following problem:

$$\min_{\substack{p_\phi:\mathcal{S}\to\Delta^{|\hat{\mathcal{R}}|-1}}} \mathop{\mathbb{E}}_{\substack{Q\sim U(\hat{\mathcal{R}})\\S\in\mathcal{D}}} \left[\mathrm{CE}\left(p_\phi(QS), \delta_Q\right)\right], \tag{5}$$

where $U(\hat{\mathcal{R}})$ denotes the uniform distribution on $\hat{\mathcal{R}}$, $\mathrm{CE}(\cdot)$ denotes the cross-entropy loss, and $\delta_Q \in \Delta^{|\hat{\mathcal{R}}|-1}$ is a one-hot vector centered at the index of $Q \in \hat{\mathcal{R}}$. While one may hope that composing the quotient regression model and this classifier yields an orienter that outputs correct orientations $\hat{\Omega}_{RS} = R\Omega_S$ regardless of its inputs' symmetries, recall that Proposition 3.1 shows such an orienter does not exist. However, the following result shows that this pipeline recovers the orientation of a shape $RS$ up to one of its rotational symmetries, which is sufficient for mapping $RS$ to $S$.

**Proposition 3.4** *Let $S \in \mathcal{S}$ be a fixed shape which is symmetric under a group of rotations $\mathcal{R}_S \subseteq \hat{\mathcal{R}} \subseteq SO(3)$, and suppose $S$ is canonically-oriented, so $\Omega_S = I$. Let $f^* : \mathcal{S} \to SO(3)$ be a solution to Equation 3, so that $f^*(RS) = RQ^*$ for some $Q^* \in \hat{\mathcal{R}}$. Finally, suppose that $p^* : \mathcal{S} \to \Delta^{|\hat{\mathcal{R}}|-1}$ solves the following problem:*

$$\min_{\substack{p:\mathcal{S}\to\Delta^{|\hat{\mathcal{R}}|-1}}} \mathop{\mathbb{E}}_{\substack{Q\sim U(\hat{\mathcal{R}})}} \left[CE\left(p(QS), \delta_Q\right)\right]. \tag{6}$$

*Then for any $R \in SO(3)$, $p^*(f^*(RS)^\top RS)$ is the uniform distribution over $\left\{(Q^*)^\top F : F \in \mathcal{R}_S\right\}$. For any $(Q^*)^\top F$ in the support of this distribution,* $\underbrace{((Q^*)^\top F)^\top}_{\textit{second-stage prediction}} \underbrace{f^*(RS)^\top}_{\textit{first-stage prediction}} RS = S$, *so using $f^*$ and $p^*$, one may recover $S$ from the arbitrarily-rotated shape $RS$.*

We prove this proposition in Appendix A.4. How does one reconcile this result with Proposition 3.1? The orientation of $(Q^*)^\top F)^\top f^*(RS)^\top RS$ is $F^\top \neq I = \Omega_S$, so this result does *not* contradict Proposition 3.1. Rather, it shows that when a shape $S$ is rotationally symmetric, one need only predict the orientation of $RS$ *up to one of its symmetries* to recover the canonically-oriented $S$. We combine these results in the following section to implement a state-of-the-art method for orientation estimation.

## 3.2 IMPLEMENTATION

Informed by our insights from Section 3.1, we now present our state-of-the-art method for orientation estimation. Our pipeline consists of two components. Our first component, which we call the

*quotient orienter*, is a neural network trained to solve Problem 3. We quotient the $L^2$ objective by $\hat{\mathcal{R}} := \mathcal{O} \subseteq SO(3)$, the *octahedral group* containing the 24 rotational symmetries of a cube. This is among the largest finite subgroups of $SO(3)$ (only the cyclic group $C_n$ for $n \geq 48$ and dihedral group $D_n$ for $n \geq 4$ can contain more subgroups), and it includes many rotational symmetries that commonly occur in real-world shapes. However, our method is general, and one may implement it with a different choice of $\hat{\mathcal{R}}$ by generating a set of rotation matrices representing the symmetries in $\hat{\mathcal{R}}$ and retraining our model. This amounts to editing a few lines of code before retraining.

Our second component, which we call the *flipper*, is a neural network trained to solve the classification problem 5. We illustrate the output of each stage of this pipeline in Figure 5. As many shapes possess multiple plausible orientations, we use conformal prediction to enable our flipper to output *adaptive prediction sets* whose size varies with the flipper model's uncertainty. We provide further implementation details below.

**Quotient orienter.**   We parametrize our quotient orienter by a DGCNN (Wang et al., 2019) operating on point clouds. To ensure that our predicted orientations lie in $SO(3)$, we follow Brégier (2021) and map model outputs from $\mathbb{R}^{3 \times 3}$ to $SO(3)$ by solving the special orthogonal Procrustes problem. We train the quotient orienter on point clouds sampled from the surfaces of meshes in Shapenet (Chang et al., 2015). As these meshes are pre-aligned to lie in canonical orientation, we fix $\Omega_S = I$ for all training shapes $S$. We provide full architecture and training details in Appendix B.

In our experiments, we observe that our quotient orienter yields accurate predictions for most input rotations $R$ but fails for a small subset of rotations. To handle this, we follow Liu et al. (2016) and employ test-time augmentation to improve our model's predictions. This consists of (1) randomly rotating the inputs $RS$ by $K$ random rotations $R_k \sim U(SO(3)), k = 1, ..., K$, (2) obtaining the quotient orienter's predictions $f_\theta(R_k RS)$ for each shape, (3) returning these predictions to the original input's orientation by computing $R_k^\top f_\theta(R_k RS)$, and (4) outputting the prediction $R_{k^*}^\top f_\theta(R_{k^*} RS)$ with the smallest average quotient distance to the remaining predictions.

**Flipper.**   We also parametrize our flipper by a DGCNN operating on point clouds. We train the flipper on point clouds sampled from the surface of Shapenet meshes by optimizing Equation 5. We draw rotations $Q \sim U(\mathcal{O})$ during training, and simulate inaccuracies in our quotient orienter's predictions by further rotating the training shapes about a randomly drawn axis by an angle uniformly drawn from $[0, 10]$ degrees. We provide full architecture and training details in Appendix B.

We also employ test-time augmentation to improve our flipper model's predictions. Similarly to the case with the quotient orienter, we (1) randomly flip the inputs by $K$ random rotations $R_k \sim \hat{\mathcal{R}} = \mathcal{O}$, (2) obtain the flipper's predictions for each shape, (3) return these predictions to the original input's orientation, and (4) output the plurality prediction.

**Adaptive prediction sets.**   Many real-world shapes have several plausible canonical orientations, even when they lack rotational symmetries. Furthermore, the flipper model may map nearly-symmetric shapes with unique canonical orientations to a uniform distribution over their near-symmetries due to factors such as insufficiently dense point clouds or the smoothness of the flipper function.

To mitigate this issue in pipelines with a human in the loop, we enable our flipper model to output *adaptive prediction sets* whose size varies with the flipper's uncertainty (Romano et al., 2020). This method uses a small *conformal calibration set* drawn from the validation data to learn a tuning parameter $\tau > 0$ that controls the size of the prediction sets. Given the flipper model's output probabilities $p_\phi(S) \in \Delta^{|\hat{\mathcal{R}}|-1}$ for some shape $S$, one sorts $p_\phi(S)$ in descending order and adds elements of $\hat{\mathcal{R}}$ to the prediction set until their total mass in $p_\phi(S)$ reaches $\tau$. Intuitively, these sets will be small when the flipper is confident in its prediction and assigns large mass to the highest-probability classes. Conversely, the sets will be large when the flipper is uncertain and assigns similar mass to most classes.

## 4   EXPERIMENTS

We now evaluate our method's performance on orientation estimation. We first follow the evaluation procedure in Pang et al. (2022) and benchmark against their "Upright-Net," which represents the current state of the art for orientation estimation. Upright-Net can only map shapes into upright

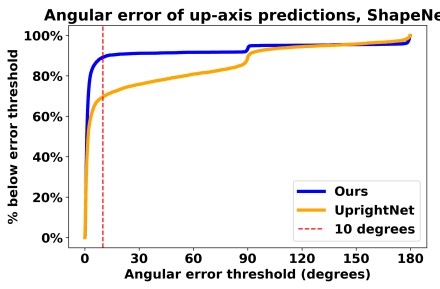 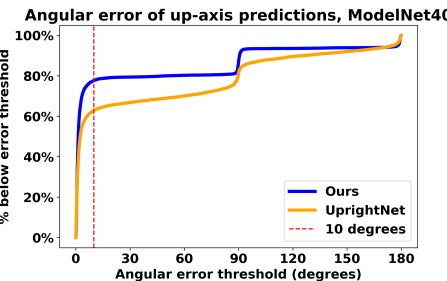

(a) Angular error histograms for Shapenet    (b) Angular error histograms for ModelNet40

Figure 6: Comparison of angular errors between the estimated and ground truth up-axis on the Shapenet validation set (left) and on ModelNet40 (right). We plot the empirical CDF of the angular errors of each model's outputs. The dashed lines indicate the $10°$ error threshold beyond which a prediction is treated as incorrect. Our algorithm's error rate is 64.6% lower than the prior state of the art.

orientation, where a shape's up axis is aligned with the $y$-axis; in contrast, our method recovers a full orientation $\Omega_S$ for each shape. We therefore follow this benchmark with an evaluation of our method on the more challenging task of full-orientation estimation. We incorporate adaptive prediction sets at this stage and demonstrate that our method reliably provides a plausible set of candidate orientations for diverse shapes unseen during training. We train and evaluate all models on Shapenet (Chang et al., 2015), as this is the largest and most diverse dataset we are aware of consisting of canonically oriented shapes. However, we report qualitative results for our method's out-of-distribution performance on Objaverse in Appendix C.2.

### 4.1 Upright orientation estimation

We construct a random 90-10 train-test split of Shapenet, draw 10k point samples from the surface of each mesh, and train our quotient orienter and flipper on all classes in the training split. We train our quotient orienter for 1919 epochs and our flipper for 3719 epochs, sampling 2k points per point cloud at each iteration and fixing a learning rate of $10^{-4}$. We also train Upright-Net with 2048 points per cloud on the same data for 969 epochs at the same learning rate, at which point the validation accuracy has plateaued. We follow the annotation procedure in Pang et al. (2022) to obtain ground truth segmentations of each point cloud into supporting base points and non-base points.

We then follow the evaluation procedure in Pang et al. (2022) to benchmark our method against their SOTA method for upright orientation estimation. We randomly rotate shapes $S$ in the validation set, use our two-stage pipeline and Upright-Net to estimate the up-axis $\omega^y_{RS}$ of each randomly rotated shape $RS$, and then measure the *angular error* $\arccos(\langle \hat{\omega}^y_{RS}, \omega^y_{RS} \rangle)$ between the estimated and ground truth up-axis. Our method's estimated up-axis is the second column of our estimated orientation matrix

Table 1: Up-axis estimation accuracy for our pipeline trained on Shapenet

| Method | Accuracy (↑) | |
|---|---|---|
| | Shapenet | ModelNet40 |
| Ours | **89.2 %** | **77.7 %** |
| Upright-Net | 69.5 % | 62.3 % |

$\hat{\Omega}_{RS}$. This metric is in fact more challenging than necessary for our method, as it treats an estimate that is correct up to a symmetry of $RS$ as a failure, even if the resulting shape is upright. We opt for this challenging metric to ensure a fair comparison against prior work.

In contrast, Upright-Net predicts a set of base points for $RS$, fits a plane to these points, and returns this plane's normal vector pointing towards the shape's center of mass. This method relies on a restrictive prior on the geometry of the input shapes and fails on shapes which do not naturally lie on a supporting base. We follow Pang et al. (2022) and define our methods' respective accuracies to be the proportion of validation meshes whose angular error is less than $10°$.

We depict the results of this benchmark in Table 1. **Our method improves on Upright-Net's up-axis estimation accuracy by nearly 20 percentage points**, corresponding to a 64.6% reduction

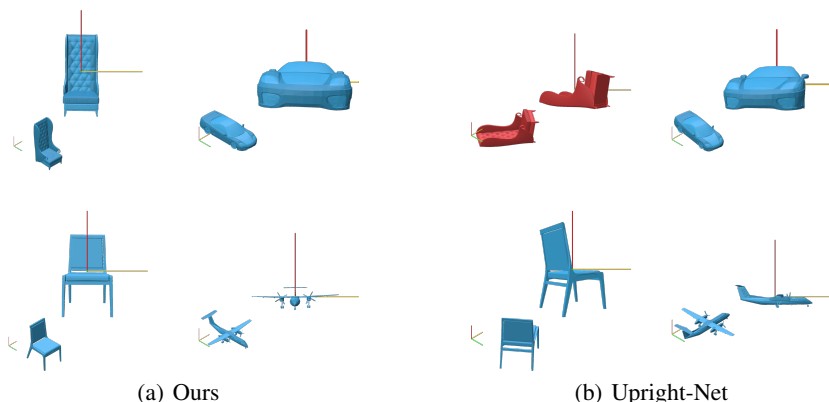

(a) Ours                                    (b) Upright-Net

Figure 7: Comparison of oriented shapes recovered from randomly rotated inputs using our algorithm (left) and Upright-Net (right). Failures are rendered in red. Our algorithm recovers correct upright and front-facing orientations for most shapes, whereas Upright-Net cannot recover front-facing orientations and fails over $2.8\times$ as often at up-axis prediction.

in the error rate relative to the previous state of the art. To provide a more comprehensive picture of our respective models' performance, we also report angular loss histograms for our model and Upright-Net in the left panel of Figure 6. Our model primarily fails by outputting orientations that are $90°$ or $180°$ away from the correct orientation, which correspond to failures of the flipper. In contrast, Upright-Net's failures are more evenly distributed across angular errors. Finally, we depict a grid of non-cherry-picked outputs of our model and Upright-Net in Figure 7 and highlight each model's failure cases in red.

We quantitatively evaluate our model's generalization by performing the same experiment on ModelNet40 (Wu et al., 2015). Both models' performances deteriorate in this setting, but our algorithm continues to substantially outperform Upright-Net. Furthermore, the right panel of Figure 6 shows that our model's failures on ModelNet40 are more heavily weighted towards flipper failures (where the angular error is close to $90°$ and $180°$). In the following section, we will show how a human in the loop can resolve these failures by choosing from a small set of candidate flips, which substantially improves our pipeline's quantitative performance.

These results demonstrate that our method significantly improves over the state of the art in upright orientation estimation. In the following section, we show that our method also successfully recovers the full orientation $\Omega_{RS}$ of a rotated shape, a more challenging task than the well-studied task of upright orientation estimation. Using our estimated orientations, we return a wide variety of shapes into canonical orientation.

### 4.2 FULL-ORIENTATION ESTIMATION

We now evaluate our method's performance on full-orientation estimation, in which we use our model's full orientation matrix $\hat{\Omega}_{RS}$ to transform an arbitrarily-rotated shape $RS$ to the canonically-oriented shape $S$. To our knowledge, our algorithm is the first to solve this task for generic shapes without requiring class information at training time or at inference time. We now record the *angular distance* between our estimated orientations $\hat{\Omega}_{RS}$ and the ground truth orientations $\Omega_{RS} = R\Omega_S = R$. This angular distance is defined as $d(\hat{\Omega}_{RS}, R) := \arccos(\frac{\mathrm{tr}R_{\mathrm{diff}}-1}{2})$, where $R_{\mathrm{diff}} := \hat{\Omega}_{RS}R^\top$.

As noted in Section 3.2, many real-world shapes have several plausible canonical orientations, and our flipper may also map nearly-symmetric shapes to a uniform distribution over their near-symmetries. In particular, while most real-world shapes have a well-defined upright orientation, their front-facing orientation is often ambiguous. To account for this, we incorporate adaptive prediction sets at this stage of our evaluation. We measure the angular distance between the estimated orientations $\hat{\Omega}_{RS}^k$ corresponding to the top $K = 4$ flips in our flipper's model distribution and the ground truth orientation $\Omega_{RS}$, and take their maximum to obtain our reported angular errors. This resolves ambiguities in a shape's front-facing direction (there are 4 possible front-facing directions for each upright orientation) and also simulates the ability of a human in the loop to choose between a small set of candidate orientations.

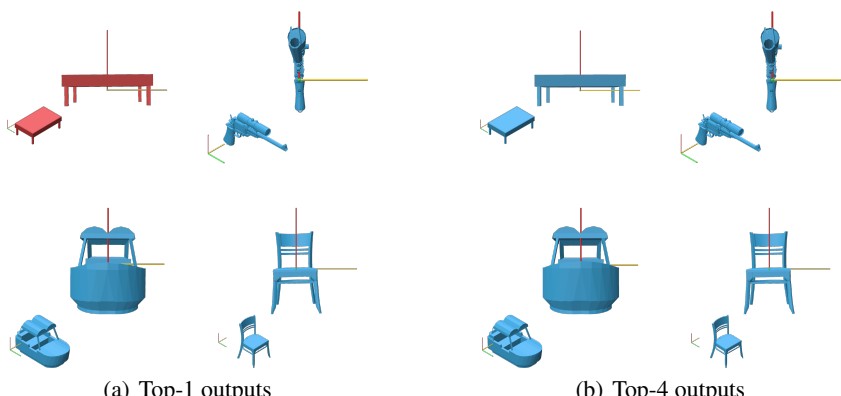

(a) Top-1 outputs          (b) Top-4 outputs

Figure 8: Comparison of oriented shapes recovered using the flipper's highest-probability flip (left) and the best flip among the top 4 classes in the flipper's model distribution (right). Our pipeline correctly infers the full orientation of most shapes, and many of its failures correspond to orientations that are acceptable in practice.

We describe our method's performance on full-orientation estimation in Table 2, where we adopt a $10°$ accuracy threshold for full rotation estimation. Our method achieves high top-4 accuracy on full-orientation estimation, but its accuracy deteriorates in the $K = 1$ case, where one can only consider the flipper's highest-probability class. This is partially attributable to ambiguities in the front-facing orientation of many shapes. To demonstrate this, the left panel of Figure 8 depicts shapes where our method's top-1 angular error is $> 10°$ in red. These failures primarily correspond to shapes that are symmetric with respect to rotations about the $y$-axis; these shapes are correctly oriented even though our method has recovered incorrect orientations.

Figure 8 compares the shapes obtained using our model's outputs when $K = 1$ (where we only consider the flipper's highest-probability predicted flip) and when $K = 4$ (where we depict the best flip among the top 4 classes in the flipper's model distribution) to the canonically-oriented shapes. We highlight the model's failure cases in red. Even in the $K = 1$ case, many of the model's failures correspond to orientations that are plausible or correct up to a symmetry of the shape. We bolster this claim with additional non-cherry-picked examples in Appendix C.2; see Figures 10, 11.

Table 2: Full-orientation estimation accuracy

| Method | Accuracy ($\uparrow$) | |
|---|---|---|
| | Top-1 | Top-4 |
| Ours | 68.3 % | 85.8 % |

Finally, in Figure 12 in Appendix C.2, we depict transformed shapes obtained by applying our method to randomly-rotated shapes from the Objaverse dataset (Deitke et al., 2023). This dataset contains highly diverse meshes of varying quality and therefore serves as a useful test case for our method's performance on out-of-distribution shapes. (As these meshes are not consistently oriented in the dataset, we cannot train on them or report meaningful error metrics.) Using our orientation pipeline, one reliably recovers shapes that are canonically-oriented up to an octahedral flip. Our flipper has greater difficulty handling out-of-distribution meshes, but predicts an acceptable flip in many cases. We expect that training our flipper on a larger dataset of oriented shapes will further improve its generalization performance.

## 5 CONCLUSION

This work introduces a state-of-the-art method for 3D orientation estimation. Whereas previous approaches can only infer upright orientations for limited classes of shapes, our method successfully recovers entire orientations for general shapes. We show that naïve regression-based approaches for orientation estimation degenerate on rotationally-symmetric shapes, which are common in practice, and develop a two-stage orientation pipeline that avoids these obstacles. Our pipeline first orients an arbitrarily rotated input shape up to an octahedral symmetry, and then predicts the octahedral symmetry that maps the first-stage output to the canonically-oriented shape. We anticipate that this factorization of geometric learning problems will be broadly applicable throughout 3D deep learning for tackling problems that are ill-posed due to the presence of symmetries. We also believe that our results can be further improved by training our quotient orienter and flipper models on larger datasets of consistently-oriented shapes as they become available.

**Reproducibility Statement.** To ensure the reproducibility of our results, we have included complete proofs of all theoretical results in Appendix A, described our implementation details in Appendix B, and uploaded our source code with the supplementary materials.

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

# A  PROOFS

## A.1  PROPOSITION 3.1

Suppose $f(RS) = R\Omega_S$ for all $R \in SO(3)$, and fix some non-identity rotation $R \in \mathcal{R}_S$ under which $S$ is symmetric. Then $RS = S$, but $f(RS) = R\Omega_S \neq \Omega_S = f(S)$, so $f(RS) \neq f(S)$ even though $RS = S$. Hence $f$ must be a one-to-many map and is therefore not a function. ∎

## A.2 PROPOSITION 3.2

The key insight is that if $f$ is a function, then $f(RS) = f(R'S)$ for all $R' \in SO(3)$ such that $RS = R'S$. Equation 2 will then drive the optimal $f^*(RS)$ to the *Euclidean mean* (Moakher, 2002) of the rotation matrices $R'$ such that $RS = R'S$. We begin by showing that these are precisely the matrices $RQ$ for $Q \in \mathcal{R}_S$.

As $\mathcal{R}_S$ is the group of symmetries of $S$, $QS = S$ for all $Q \in \mathcal{R}_S$. Given some rotation $R \in SO(3)$, left-multiplying by $R$ then yields $RQS = RS$ for all $Q \in \mathcal{R}_S$. This relationship also holds in reverse: If $RS = R'S$ for $R, R' \in SO(3)$, then $R' = RQ$ for some $Q \in \mathcal{R}_S$. To see this, note that if $RS = R'S$, then $S = R^\top R'S$ and hence $R^\top R' \in \mathcal{R}_S$. Consequently, $R' = R(R^\top R') = RQ$ for $Q := R^\top R' \in \mathcal{R}_S$. It follows that:

$$\{R' \in SO(3) : RS = R'S\} = \{RQ : Q \in \mathcal{R}_S\}.$$

We can therefore write a solution to Equation 2 evaluated at $RS$ as follows:

$$\begin{aligned}
f^*(RS) &= \operatorname*{argmin}_{R^* \in SO(3)} \mathbb{E}_{R' \in SO(3):RS=R'S} \left[ \|R^* - R'\Omega_S\|_F^2 \right] \\
&= \operatorname*{argmin}_{R^* \in SO(3)} \mathbb{E}_{RQ:Q \in \mathcal{R}_S} \left[ \|R^* - RQ\Omega_S\|_F^2 \right] \\
&= \operatorname*{argmin}_{R^* \in SO(3)} \mathbb{E}_{Q \in U(\mathcal{R}_S)} \left[ \|R^* - RQ\Omega_S\|_F^2 \right] \\
&= \operatorname*{argmin}_{R^* \in SO(3)} \frac{1}{|\mathcal{R}_S|} \sum_{Q \in \mathcal{R}_S} \|R^* - RQ\Omega_S\|_F^2.
\end{aligned}$$

This is the Euclidean mean of the matrices $RQ\Omega_S$ as defined in Moakher (2002). Proposition 3.3 in the same reference states that the solution to this problem is found by computing the arithmetic mean $\frac{1}{|\mathcal{R}_S|} \sum_{Q \in \mathcal{R}_S} RQ\Omega_S$ and then orthogonally projecting this onto $SO(3)$. In particular,

$$f^*(RS) = \operatorname{proj}_{SO(3)} \left[ \frac{1}{|\mathcal{R}_S|} \sum_{Q \in \mathcal{R}_S} RQ\Omega_S \right] \neq R\Omega_S.$$

Hence $L^2$ regression fails to learn the orientation $\Omega_S$ of a shape $S \in \mathcal{S}$ that possesses a non-trivial set of rotational symmetries $\mathcal{R}_S$. ∎

## A.3 PROPOSITION 3.3

We begin by defining an equivalence relation over $SO(3)$. Given two rotations $R_1, R_2 \in SO(3)$, we call $R_1, R_2$ equivalent and write $R_1 \sim R_2$ if there exists some $Q \in \hat{\mathcal{R}}$ such that $R_2 = R_1 Q$. We verify that this is an equivalence relation:

**Reflexivity:** $I \in \hat{\mathcal{R}}$ since $\hat{\mathcal{R}}$ is a group and $R_1 = R_1 I$, so $R_1 \sim R_1$.

**Symmetry:** Suppose $R_1 \sim R_2$. Then $R_2 = R_1 Q$ for some $Q \in \hat{\mathcal{R}}$. As $\hat{\mathcal{R}}$ is a group, $R^\top = R^{-1} \in \hat{\mathcal{R}}$ as well, and $R_2 Q^\top = R_1$, so $R_2 \sim R_1$.

**Transitivity:** Suppose $R_1 \sim R_2$ and $R_2 \sim R_3$. Then there are $Q, Q' \in \hat{\mathcal{R}}$ such that $R_2 = R_1 Q$ and $R_3 = R_2 Q'$. Hence $R_3 = R_2 Q' = R_1 Q Q'$, and as $\hat{\mathcal{R}}$ is a group, $QQ' \in \hat{\mathcal{R}}$. We conclude that $R_1 \sim R_3$.

This confirms that $\sim$ is a valid equivalence relation. Using this equivalence relation, we partition $SO(3)$ into equivalence classes, choose a unique representative for each class, and use $[R] \in SO(3)/\sim$ to denote the unique representative for the equivalence class containing $R \in SO(3)$.

We then use this map to define a candidate solution to Equation 4 over the space of rotated shapes $\{RS : R \in SO(3)\}$ as $f^*(RS) := [R]\Omega_S$. We will first verify that this defines a valid function (i.e. that $f^*$ is not one-to-many), and then show that it attains a loss value of 0 in Equation 4.

We first show that $f^*$ defines a valid function. To do so, we must show that if $R_1S = R_2S$, then $f^*(R_1S) = f^*(R_2S)$. To this end, suppose that $R_1S = R_2S$. Then $S = R_1^\top R_2 S$, so $Q := R_1^\top R_2 \in \mathcal{R}_S \subseteq \hat{\mathcal{R}}$. It follows that $R_2 = R_1 R_1^\top R_2 = R_1 Q$ for some $Q \in \hat{\mathcal{R}}$, so $R_1 \sim R_2$. Since $R_1 \sim R_2$, $[R_1] = [R_2]$ and so $f^*(R_1S) = [R_1]\Omega_S = [R_2]\Omega_S = f^*(R_2S)$. This shows that $f^*$ defines a valid function.

We now show that $f^*$ attains a loss value of 0 in Equation 4. For any $R \in SO(3)$, we have:

$$\min_{Q \in \hat{\mathcal{R}}} \|f(RS) - RQ\Omega_S\|_F^2 = \min_{Q \in \hat{\mathcal{R}}} \|[R]\Omega_S - RQ\Omega_S\|_F^2.$$

But clearly $R \sim [R]$, so there exists some $Q^* \in \hat{\mathcal{R}}$ such that $[R] = RQ^*$. Hence

$$\min_{Q \in \hat{\mathcal{R}}} \|[R]\Omega_S - RQ\Omega_S\|_F^2 = 0,$$

and as this reasoning holds for any $R \in SO(3)$, it follows that

$$\mathbb{E}_{R \sim U(SO(3))} \left[\min_{Q \in \hat{\mathcal{R}}} \|f(RS) - RQ\Omega_s\|_F^2\right] = 0.$$

We conclude that $f^*$ is a minimizer of Equation 4. Furthermore, $f^*(S) = [I]\Omega_S = Q^*\Omega_S$ for some $Q^* \in \hat{\mathcal{R}}$, which completes the proof of the proposition. ∎

### A.4 PROPOSITION 3.4

If $F \in \mathcal{R}_S$, then $FS = S$, so $QFS = QS$ for any other rotation $Q \in SO(3)$ and $\{QFS : F \in \mathcal{R}_S\}$ contains the symmetries of the rotated shape $QS$. The optimal solution $p^*$ to Equation 6 maps a rotated shape $QS$ (where $Q \in \hat{\mathcal{R}}$) to the empirical distribution of the targets $Q \in \hat{\mathcal{R}}$ conditional on a shape $QS$. But if $QFS = QS$ for all $F \in \mathcal{R}_S$, then this is the uniform distribution over the set $\{QF : F \in \mathcal{R}_S\}$.

Since $f^*(RS) = RQ^*$ for some $Q^* \in \hat{\mathcal{R}}$, $f^*(RS)^\top RS = (Q^*)^\top S$, and applying the general result from above, we conclude that $p^*(f^*(RS)^\top RS) = p^*((Q^*)^\top S)$ is the uniform distribution over the set $\{(Q^*)^\top F : F \in \mathcal{R}_S\}$.

For any $(Q^*)^\top F$, one then computes $((Q^*)^\top F)^\top f^*(RS)^\top RS = F^\top S$. But as $\mathcal{R}_S$ is a group, $F^\top \in \mathcal{R}_S$ whenever $F$ is, so $F^\top S = S$ and we conclude that $((Q^*)^\top F)^\top f^*(RS)^\top RS = S$. ∎

## B IMPLEMENTATION DETAILS

### B.1 QUOTIENT ORIENTER

We parametrize our quotient orienter by a DGCNN and use the author's Pytorch implementation (Wang et al., 2019) with 1024-dimensional embeddings, $k = 20$ neighbors for the EdgeConv layers, and a dropout probability of 0.5. Our DGCNN outputs unstructured $3 \times 3$ matrices, which we then project onto $SO(3)$ by solving a special orthogonal Procrustes problem; we use the roma package (Brégier, 2021) to efficiently compute this projection.

We train our quotient orienter on point clouds consisting of 10k surface samples from Shapenet meshes. We subsample 2k points per training iteration and pass batches of 48 point clouds per iteration. We train the quotient orienter for 1919 epochs at a learning rate of $10^{-4}$.

For test-time augmentation, we (1) randomly rotate the inputs $RS$ by $K$ random rotations $R_k \sim U(SO(3))$, $k = 1, ..., K$, (2) obtain the quotient orienter's predictions $f_\theta(R_kRS)$ for each shape,

(3) return these predictions to the original input's orientation by computing $R_k^\top f_\theta(R_k R S)$, and (4) output the prediction $R_{k^*}^\top f_\theta(R_{k^*} R S)$ with the smallest average quotient distance to the remaining predictions.

## B.2 FLIPPER

We parametrize our flipper by a DGCNN and use the author's Pytorch implementation (Wang et al., 2019) with 1024-dimensional embeddings, $k = 20$ neighbors for the EdgeConv layers, and a dropout probability of 0.5. Our flipper outputs 24-dimensional logits, as we quotient our first-stage regression problem by the octahedral group, which contains the 24 rotational symmetries of a cube.

We train our flipper on point clouds consisting of 10k surface samples from Shapenet meshes. We subsample 2k points per training iteration and pass batches of 48 point clouds per iteration. We train the quotient orienter for 3719 epochs at a learning rate of $10^{-4}$. We draw rotations $Q \in U(\mathcal{O})$ during training, and simulate inaccuracies in our quotient orienter's predictions by further rotating the training shapes about a randomly drawn axis by an angle uniformly drawn from $[0, 10]$ degrees.

We also employ test-time augmentation to improve our flipper model's predictions. Similarly to the case with the quotient orienter, we (1) randomly flip the inputs by $K$ random rotations $R_k \sim \hat{\mathcal{R}} = \mathcal{O}$, (2) obtain the flipper's predictions for each shape, (3) return these predictions to the original input's orientation, and (4) output the plurality prediction.

## B.3 ADAPTIVE PREDICTION SETS

We implement adaptive prediction sets following the method in Angelopoulos et al. (2020) with their regularization parameter $\lambda$ set to 0. To calibrate our *conformal flipper*, we first draw a subset of the validation set (the *calibration set*), apply a random octahedral flip $Q \sim U(\mathcal{O})$ to each calibration shape, and then pass each flipped shape $QS$ through the trained flipper to obtain class probabilities $p_\phi(QS) \in \Delta^2 3$. The *calibration score* for a shape $S$ is the sum of the model's class probabilities $p(QS)_i$ ranked in descending order, up to and including the true class $i^*$ corresponding to the ground truth flip $Q$. We fix a confidence level $1 - \alpha$ and return the $(1 - \alpha)$-th quantile $\tau$ of the calibration scores for each shape in the calibration set. In general, smaller values of $1 - \alpha$ lead to smaller values of $\tau$, which ultimately results in smaller prediction sets at inference time, whereas large values of $1 - \alpha$ lead to larger prediction sets at inference time but with stronger guarantees that these sets include the true flip.

At inference time, we first obtain the flipper model's output probabilities $p_\phi(S) \in \Delta^{|\hat{\mathcal{R}}|-1}$ for some shape $S$, then sort $p_\phi(S)$ in descending order and add elements of $\hat{\mathcal{R}}$ to the prediction set until their total mass in $p_\phi(S)$ reaches $\tau$. Intuitively, these sets will be small when the flipper is confident in its prediction and assigns large mass to the highest-probability classes. Conversely, the sets will be large when the flipper is uncertain and assigns similar mass to most classes.

# C ADDITIONAL RESULTS

## C.1 PAIRWISE ANGULAR ERRORS

In this section, we replicate the shape alignment experiment from Zhou et al. (2022, Section 4.2) using our two-stage shape orientation pipeline. Following Zhou et al. (2022), we apply random azimuthal rotations to the airplane meshes in the Shapenet validation set and use our pipeline to predict the orientation of each randomly-rotated airplane mesh. We then compute the pairwise angular distance between each pair of ground truth orientations $(\Omega_i, \Omega_j)$ and predicted rotations $(\hat{\Omega}_i, \hat{\Omega}_j)$ using a generalization of the alignment metric proposed by Averkiou et al. (2016); this is the metric employed by Zhou et al. (2022) for their shape alignment experiments. The formula for this metric is as follows:

$$d\left((\Omega_i, \Omega_j), (\hat{\Omega}_i, \hat{\Omega}_j)\right) := \arccos\left(\frac{\mathrm{tr} R_{\mathrm{diff}}^{ij} - 1}{2}\right),$$

where the pairwise difference rotation $R_{\mathrm{diff}}^{ij}$ is defined as $R_{\mathrm{diff}}^{ij} = (\Omega_i \Omega_j)(\hat{\Omega}_i \hat{\Omega}_j))^\top$.

Figure 9: Empirical CDF of our model's pairwise angular errors on airplane meshes. 89.1% of the pairwise angular errors are under 10 degrees (red dashed line).

Following Zhou et al. (2022), we depict the empirical CDF of our model's pairwise angular errors on the airplanes meshes in the validation set. 89.1% of the pairwise angular errors are under 10 degrees, which compares favorably to the roughly 80% reported by Zhou et al. (2022).

## C.2 ADDITIONAL FIGURES

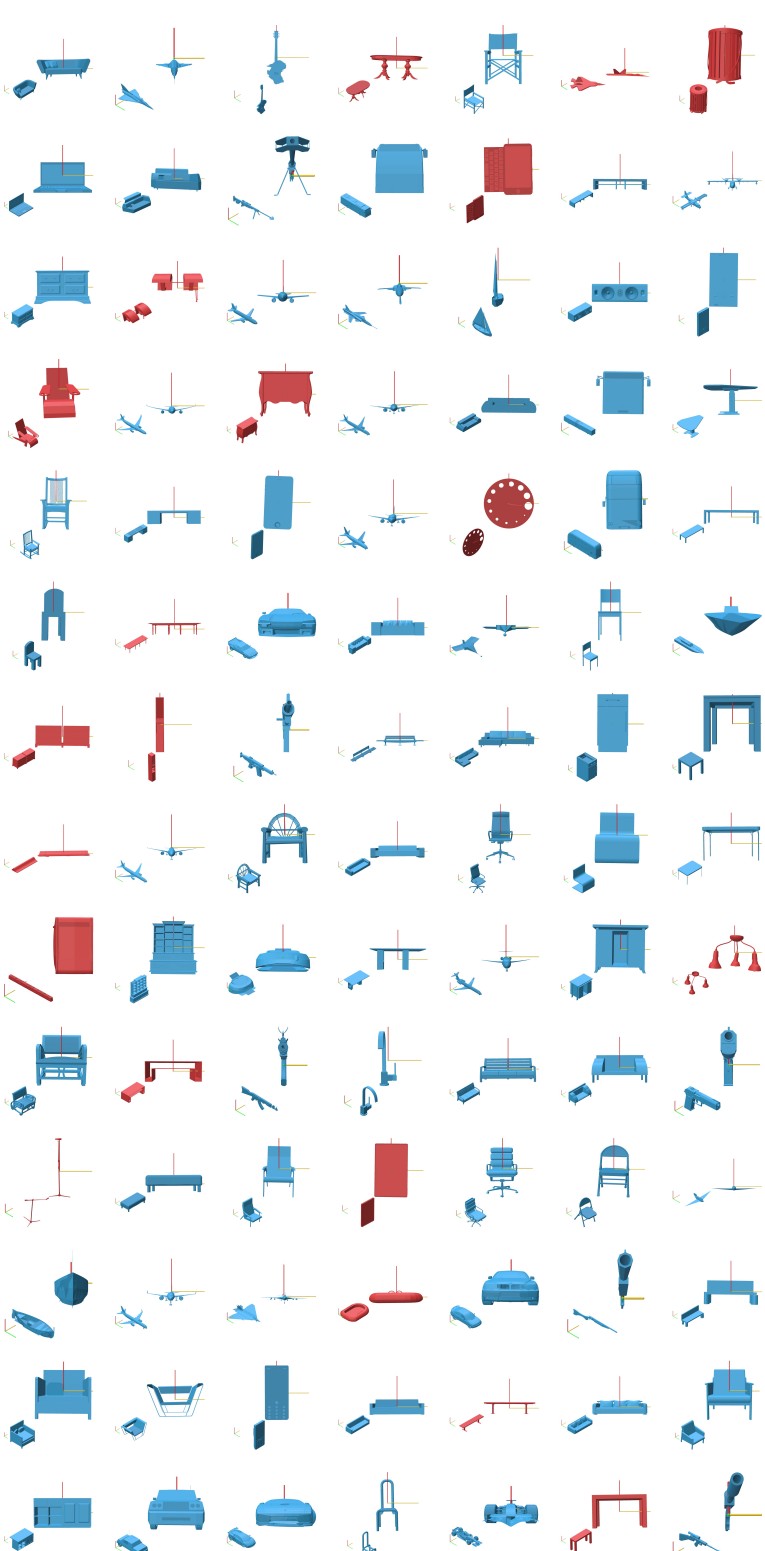

Figure 10: A grid of our model's top-1 outputs given randomly rotated, non-cherrypicked meshes from Shapenet. We depict a front view of our model's recovered shapes along with an inset depicting an isometric view and highlight failure cases in red. Even when our model fails to recover the correct orientation, the recovered shape is often acceptable in practice.

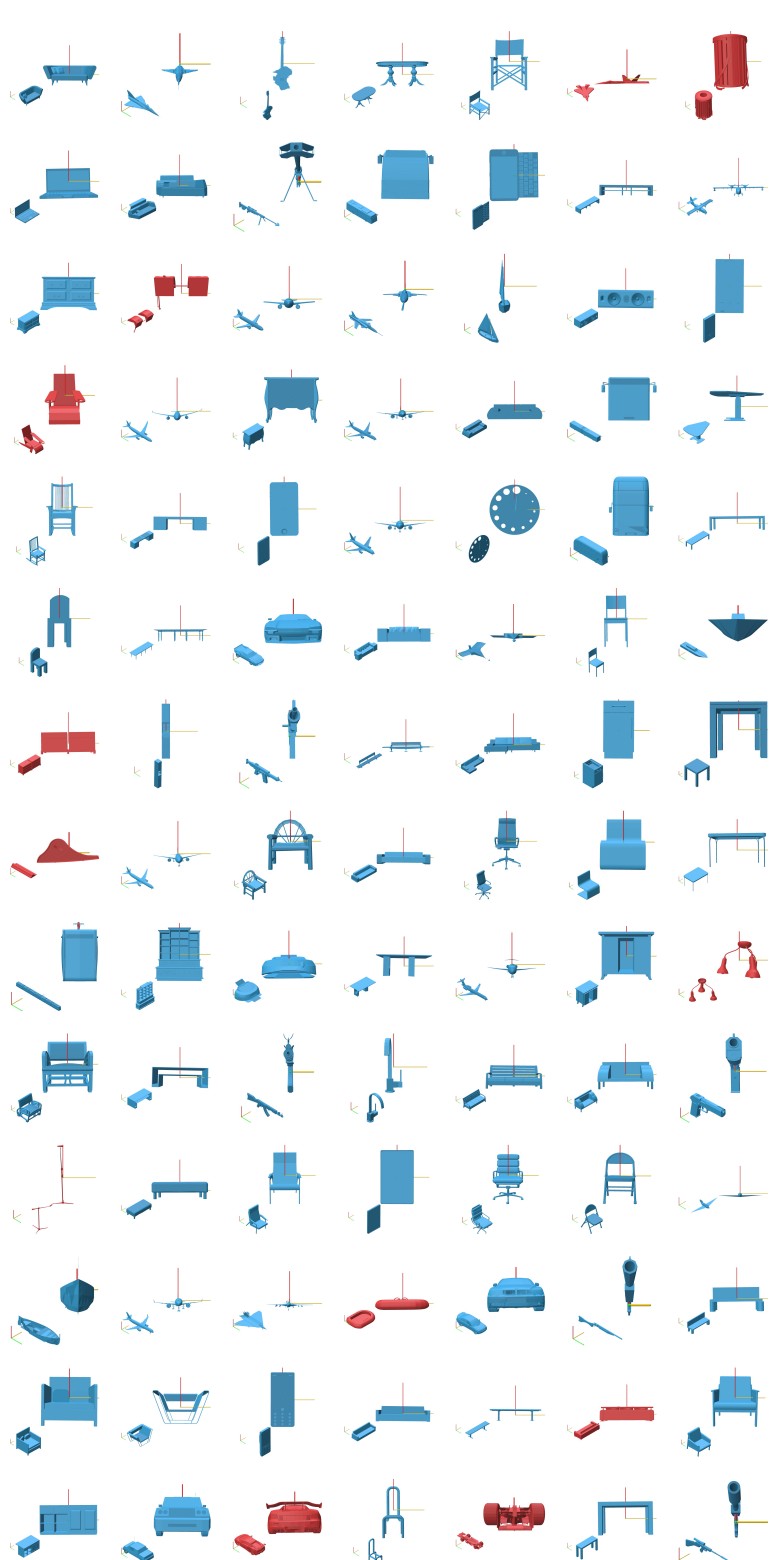

Figure 11: A grid of our model's top-4 outputs given randomly rotated, non-cherrypicked meshes from Shapenet. We depict a front view of our model's recovered shapes along with an inset depicting an isometric view and highlight failure cases in red.

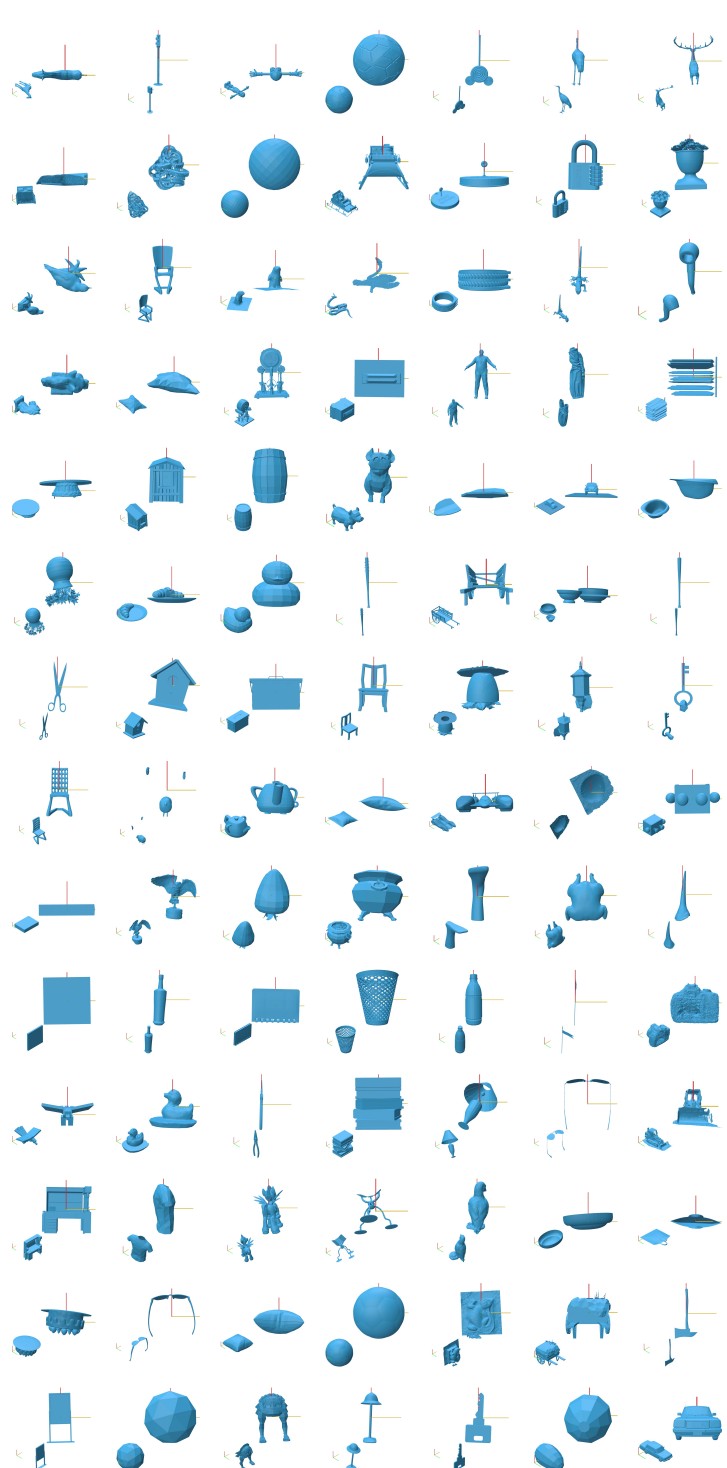

Figure 12: A grid of our model's top-1 outputs given randomly rotated, non-cherrypicked meshes from Objaverse. We depict a front view of our model's recovered shapes along with an inset depicting an isometric view. Our quotient regressor consistently succeeds on out of distribution meshes, as most of our pipeline's outputs are correctly oriented up to a cube flip. Our flipper has greater difficulty generalizing, but predicts an acceptable flip in many cases. We expect that training on a larger and more diverse dataset of oriented shapes will improve our flipper's generalization performance.

