# OpenReview forum: "Orient Anything"
_ICLR.cc/2025/Conference — Submitted to ICLR 2025_

### Official Review · Reviewer_acsD · 2024-10-29

**Soundness:** 3
**Presentation:** 2
**Contribution:** 2
**Rating:** 5
**Confidence:** 4

**Summary:**

The paper addresses the problem of estimating the orientation of rigid 3D shapes. The paper starts from the fact that, theoretically, it is not possible to have a function that, for any arbitrarily oriented shape with non-trivial rotational symmetries, predicts the right orientation (output is not unique, so such a map cannot be a function). The paper also shows that L2 objectives are not suitable for recovering the solution by optimization.

Hence, the paper proposes to decompose the problem in two parts:
1) The Quotient Orienter: a neural network (DGCNN) that solves for the rotation with an L2 objective quotient by the octahedral group.
2) The Flipper: a neural network (DGCNN) trained to solve a classification problem to recover one of the 24 octahedral flips, returning the canonical orientation.

The approach is compared against one baseline, which shows significantly better accuracy on ShapeNet and ModelNet.

**Strengths:**

- The paper addresses a valuable problem that could potentially impact many different research areas. Symmetries have been widely studied in computer graphics and computer vision and have mainly focused on the recent Geometric Deep Learning theory.
- While previous approaches focus on learning orientation for a single object class, the proposed approach aims to provide a unified model that works on all the object classes at once.
- The theoretical discussion of the paper seems reasonable, and the code is provided in supplementary. This could be an instrumental tool for several downstream tasks, and results are probably easy to replicate.

**Weaknesses:**

1) My main concern lies in the experimental evaluation of the approach. The method compares against Upright-net, Shapenet, and ModelNet. Although I consider favorably the use of the entire ShapeNet dataset, I generally think more comparisons would be informative. For example, ART ("Adjoint Rigid Transform Network: Task-conditioned Alignment of 3D Shapes", Zhou et al., 3DV 2022) seems relevant but not discussed in the paper and could serve as a baseline for the category of "canonical alignment" methods. Notice that ART also showcases results on humans. This latter is also an interesting domain, and I would be interested to know the authors' opinions on whether their method would suit this case. Similarly, I evaluate positively that the method has been tested on Objaverse, but the few qualitative examples do not help to understand the actual flexibility of the method.
2) Although I enjoyed reading the method, I think there is room to improve the presentation. The most severe problem is probably the choice of qualitative visualization, which is difficult to read and compare. I would suggest visualizing fewer examples but bigger ones with clearer axe orientations, as well as pairing them with the GT for comparison. The method is also quite extensive. In the end, the conclusion is relatively straightforward: although the problem of recovering an orientation cannot be formulated as a function, it is actually enough to restrict to predict one valid solution (and actually, the only doable things, since the solution in the same equivalence class are indistinguishable). As a consequence, this also makes the problem tractable with an L2 optimization. This is reasonable and could be expressed in a much more direct way. Finally, Figure 6 seems a bit deceptive due to the logarithmic scale. The number of valid shapes (bars on the left of the dashed lines) seems similar between the two methods, but the quantitative metrics indicate differently. The larger difference between the blue and red bars is often in the scale of 10¹, while some small differences in the bins of 90 and 180 degrees are actually in the 10³. I suggest removing the logarithmic scale or providing a cumulative curve visualization (e.g., on the y-axis, the percentage of shapes below or equal to the x-axis angular error).
3) The papers do not offer any ablation nor analysis of the method. This would be important to validate intuition and guide future work. For example:\
(a) how would the Flipper perform without the quotient oriented or with an oriental that does not quotient the space? \
(b) How does the model scale with the number of different data and classes? Is it beneficial to consider all the classes together, or would it be better to train specialized models for every class? \
(c) What is the computational time in terms of inference and training? \
(d) Figure 6 seems to suggest that the error of the proposed method focuses on some sharp bins (e.g., 90 and 180 degrees, but also 45 and 135 for ModelNet). Is it due to the octahedral nature of the Flipper classifier? \
(e) Why use octahedral and not icosahedral symmetries?

I think these are all aspects that would be worth further investigation.

**Questions:**

Please, refer to the questions listed in points (1) and (3) of weaknesses.

Minor:
1) Line 335:"We observe that our quotient orienter [...] fails for a small subset of rotations". Could you elaborate more on this? Have you noticed a recurrent pattern in the failure mode?
2) "Our method improves on Upright-Net’s up-axis estimation accuracy by nearly 20 percentage points, corresponding to a 64.6% reduction in the error rate relative to the previous state of the art.". Could you elaborate a bit more about what are the terms of this "reduction"?

---

> ### Author Response · Authors · 2024-11-19
> **Rebuttal to Reviewer acsD (Part 1/3)**
>
> **Part 1/3**
>
> Thank you for your thoughtful review. **Our primary contribution in “Orient Anything” is a shape orientation pipeline that surpasses the previous state of the art by a wide margin on well-established benchmarks in this domain.** In particular, our pipeline outperforms the previous SOTA (Upright-Net) on up-axis estimation by nearly **20 percentage points**, which corresponds to a **64.6% reduction in error rate.** Furthermore, our method is able to reliably return randomly-rotated shapes drawn from all classes in ShapeNet to upright and front-facing orientation; to our knowledge, our method is the first to solve this task. We are able to do this thanks to a novel decomposition of the shape orientation problem into quotient regression and classification, which we support with rigorous theory proving that our method can recover orientations up to symmetries in the input shapes. We contrast this with a naive regression strategy, which fails on rotationally-symmetric shapes; these results shed light on why previous work like Liu et al.’s “Upright orientation of 3D shapes with convolutional networks” have had to resort to strategies such as training $n$ different orienters via regression.
>
> **We have uploaded a revised manuscript**, with new text highlighted in blue. We individually address your concerns below:
>
> *For example, ART ("Adjoint Rigid Transform Network: Task-conditioned Alignment of 3D Shapes", Zhou et al., 3DV 2022) seems relevant but not discussed in the paper and could serve as a baseline for the category of "canonical alignment" methods.*
>
> Thank you for pointing out this reference. We have included it in the “canonical alignment” subsection of our related work in the revised manuscript. These works only attempt to learn a consistent orientation within each class, but this orientation is not consistent across classes and is not generally aligned with the coordinate axes. In contrast, we tackle the more challenging task of inferring a canonical orientation that is consistent across all objects.
>
> We have nonetheless replicated the shape alignment experiment in Section 4.2 of Zhou et al. (2022) to the best of our ability in Appendix C.1 of the revised manuscript. 89.1% of our method’s pairwise angular errors are less than 10 degrees, which significantly exceeds the roughly 80% achieved by Zhou et al (2022). (Their manuscript and Github repo do not provide fine-grained results beyond Figure 7 in their manuscript.) Furthermore, note that our method outputs airplanes that are both upright and front-facing, whereas Zhou et al (2022) only attempts to bring the airplanes into a consistent orientation, which may not be upright or front-facing.
>
> *Similarly, I evaluate positively that the method has been tested on Objaverse, but the few qualitative examples do not help to understand the actual flexibility of the method.*
>
> We benchmarked our method’s performance on ShapeNet because it is the largest and most diverse dataset we are aware of whose shapes are canonically oriented. We cannot quantitatively benchmark our method on Objaverse because its shapes are in arbitrary orientations; we consequently lack a consistent ground truth orientation for each shape against which to compare our method’s inferred orientations. Given the current level of interest in 3D deep learning, we anticipate that larger datasets of canonically oriented shapes will soon be released, giving us the opportunity to train and benchmark our pipeline on larger datasets. We have emphasized this point in lines 400-402 of the revised manuscript.
>
> *I would suggest visualizing fewer examples but bigger ones with clearer axe orientations, as well as pairing them with the GT for comparison.*
>
> Thank you for this helpful suggestion. We have updated Figures 7 and 8 in the revised manuscript to include fewer shapes and have increased the size of the axes. We visualize meshes in blue if their orientation is accurate within 10 degrees, and in red otherwise. As in our original manuscript, we include larger grids of model outputs in Figures 10 and 11 in Appendix C.2.
>
> *Finally, Figure 6 seems a bit deceptive due to the logarithmic scale. The number of valid shapes (bars on the left of the dashed lines) seems similar between the two methods, but the quantitative metrics indicate differently. The larger difference between the blue and red bars is often in the scale of 10¹, while some small differences in the bins of 90 and 180 degrees are actually in the 10³. I suggest removing the logarithmic scale or providing a cumulative curve visualization (e.g., on the y-axis, the percentage of shapes below or equal to the x-axis angular error).*
>
> Thank you for this helpful suggestion; we have updated Figure 6 in our revised manuscript in response to it. We now follow Zhou et al. (2022) and plot the empirical CDF of the angular errors of each model’s outputs.
>
> **Part 2/3 below.**

---

> ### Author Response · Authors · 2024-11-19
> **Rebuttal to Reviewer acsD (Part 2/3)**
>
> **Part 2/3**
>
> *(a) how would the Flipper perform without the quotient oriented or with an oriental that does not quotient the space?*
>
> The flipper’s role in our pipeline is to predict one of 24 octahedral symmetries given the output of the quotient orienter. If both models are trained to optimality, the composition of the quotient orienter and flipper’s predicted rotations will return the shape to canonical orientation up to an octahedral symmetry. The flipper’s training set consists of canonically-oriented shapes to which we apply an octahedral symmetry and a small amount of rotational noise to reflect small inaccuracies in the quotient orienter’s output.
>
> We do not expect the flipper to be able to orient a shape on its own, as our pipeline’s inputs consist of arbitrarily-oriented shapes and the flipper can only predict octahedral symmetries. If preceded by an orienter trained without quotienting, we would not expect the flipper to further improve performance. This is because the un-quotiented L2 loss may have minima that do not correspond to the canonical orientation up to an octahedral symmetry; we give an example on lines 218-234.
>
> *(c) What is the computational time in terms of inference and training?*
>
> The inference time for our pipeline with the hyperparameters used to generate the results in our paper is less than one second per mesh on an A10 GPU. The training time for the quotient orienter is 1 week, and the training time for the flipper is just under 2 weeks, both on a single GeForce RTX 3090 GPU.
>
> *(d) Figure 6 seems to suggest that the error of the proposed method focuses on some sharp bins (e.g., 90 and 180 degrees, but also 45 and 135 for ModelNet). Is it due to the octahedral nature of the Flipper classifier?*
>
> The 90 and 180 degree errors are indeed caused by flipper failures, which account for the majority of our pipeline’s failures. We conjecture that the modes around 45 and 135 degrees are caused by the presence of shapes with symmetries that are not contained in the octahedral group. Such cases are unavoidable for any choice of symmetry group $\hat{\mathcal{R}}$, but our pipeline achieves very strong overall performance by quotienting by the octahedral group. Our method is general and can easily be adapted to different choices of $\hat{\mathcal{R}}$ depending on the expected characteristics of the data.
>
> *(e) Why use octahedral and not icosahedral symmetries?*
>
> Quotienting by the icosahedral group is a reasonable option. However, as it is not a supergroup of the octahedral group, it is not strictly superior to the octahedral group for our purposes. We quotient by the octahedral group because we expect real world shapes to include octahedral symmetries more often than icosahedral symmetries.
>
> That being said, our method can easily be adapted to any symmetry group. Because our theory is general, our theorems hold for any subgroup $\hat{\mathcal{R}}$ of $SO(3)$ containing the symmetries of the input shapes. In practice, implementing our method with a different choice of $\hat{\mathcal{R}}$ is as simple as generating a set of rotation matrices representing the symmetries in $\hat{\mathcal{R}}$ and retraining our model. This can be done in a few lines of code. Exposing this design choice in a user-friendly manner is among our paper’s novel contributions and enables our method to be adapted to specialized datasets. We have elaborated on this point in Section 3.2 of the revised manuscript (see lines 328-331).
>
> *Line 335:"We observe that our quotient orienter [...] fails for a small subset of rotations". Could you elaborate more on this? Have you noticed a recurrent pattern in the failure mode?*
>
> We have noted that the quotient orienter succeeds on the vast majority of rotations of the input shapes and fails on a small proportion of rotations, but have not noticed a recurrent pattern in which rotations it fails on. However, this failure pattern is especially amenable to test-time augmentation. Intuitively, the strategy is to randomly rotate the input shape several times and obtain the quotient orienter’s predictions for each of these inputs. The failure pattern we noticed implies that the quotient orienter will succeed on these randomly-rotated inputs with high probability, and we can aggregate this information to obtain a more accurate prediction. We provide more details on this augmentation in Appendix B.2.
>
> **Part 3/3 below.**

---

> ### Author Response · Authors · 2024-11-19
> **Rebuttal to Reviewer acsD (Part 3/3)**
>
> **Part 3/3**
>
> *"Our method improves on Upright-Net’s up-axis estimation accuracy by nearly 20 percentage points, corresponding to a 64.6% reduction in the error rate relative to the previous state of the art.". Could you elaborate a bit more about what are the terms of this "reduction"?*
>
> We follow Upright-Net and define a correct prediction for a rotated shape $RS$, with $S$ in canonical orientation, as an estimate of the up-axis whose angular distance from the true up-axis of $RS$ is <10 degrees. This metric is in fact more challenging than necessary for our method, as it treats an estimate that is correct up to a symmetry of $RS$ as a failure even though the resulting shape would be upright. We chose this challenging metric to ensure a fair comparison against prior work. We include this point in lines 423-425 of the revised manuscript. The error rate for each method is then the proportion of incorrect predictions on the validation set. Our method’s error rate is 64.6% lower than Upright-Net’s error rate when trained and evaluated on the same datasets.
>
> **Conclusion.**
>
> We thank you for your review of our paper and hope that we have addressed your concerns. If you have any remaining questions, we would be pleased to continue this conversation during the discussion phase. Otherwise, **we respectfully ask that you raise your score for this submission.**

---

> > ### Comment · Reviewer_acsD · 2024-11-22
> > **Rebuttal answer**
> >
> > I thank the authors for their reply. I appreciate their effort in clarifying my concerns. Here, I follow up on some of the topics discussed.
> >
> > 1) Canonicalization: I see that previous works mainly solve for single class orientation, while this work aims to address multiple classes at once. As mentioned by the authors, this poses a challenge for evaluation since (a) inter-class alignment is highly ambiguous; (b) there are no dataset or evaluation protocols well defined for this task. In Upright-Net paper, they address this ambiguity by defining a common orientation in terms of "base plane". In this paper, the evaluation is partially based on upright orientation estimation to compare with Upright-Net, but here, "orientation" is meant in a broader (360°) meaning, so canonicalization baselines also seem natural. This ambiguity might be solved by user input (user intervention is also suggested in Line 139), but this is generally true even when different networks are trained for every class (while they solve a way simpler task, and so should get better class-specific performance). A comparison with class-specific methods trained on the same data and hardware would be informative to show the benefits of having a single inter-class module.
> >
> > 2) Objaverse: I see the problem of obtaining a quantitative evaluation, but I would still suggest including further qualitative results for this dataset.
> >
> > 3) Presentation: I do not think the proposed solution addressed the visualization problem, and I still find the figures a bit of low quality. E.g., reorganizing the shapes in line and using just one aerial point of view per shape would provide the same information but give more space to make the figures clearer -- but generally, I think a deeper revision of this would be useful to make the paper clearer. Also, the content added in this new revision has not been properly formatted (e.g., Appendix C.1, Figure 9 caption is "Caption"), and not all the new additions have been marked in blue, as mentioned in the rebuttal.
> >
> > 4) Analysis\Ablation: I thank the author for their feedback on the individual points. I still believe that an actual quantiative ablation of some of these aspects would be informative and ground some of the statements, e.g., "real world shapes to include octahedral symmetries more often than icosahedral symmetries."; but, since the method considers multiclasses, maybe a wider symmetric group can help to better cover the different cases of classes. Also, some of these points seem to be easy to implement, requiring minor modifications while providing useful insights for the reader
> >
> > 5) Computation time: I have been quite surprised by the training time, which is quite a lot (in total, around 3 weeks). Can the authors provide the reasons behind this? What are the main bottlenecks in the training? Is it due to the slow convergence of the network, or are forward/backward passes actually slow?
> >
> > I am looking forward to hearing their (and other reviewers') opinions on these and other points raised in the reviews.
> >
> > Again, I thank the authors for their time and work.

---

### Official Review · Reviewer_XS2K · 2024-11-03

**Soundness:** 3
**Presentation:** 3
**Contribution:** 2
**Rating:** 5
**Confidence:** 3

**Summary:**

The paper presents an approach to predict the canonical 3D pose of an arbitrarily posed 3D point cloud of an object. The proposed two-stage pipeline first predicts an upright pose, using a DGCNN backbone to regress a rotation matrix from a point cloud. Then it performs a 24-way classification to resolve the remaining degree of freedom. Compared to Upright-Net (2022), it is able to find the upward orientation better on ShapeNet and ModelNet40. Top-1 and top-4 accuracy is reported for full orientation prediction.

**Strengths:**

Clarity: easy to read and the problem definition is clear.

**Weaknesses:**

In my view, the technical rigor does not meet the conference's standards. I appreciated the careful explanation and presentation, but spending multiple pages on why a direct L2 regression of rotational symmetry doesn't work seems unnecessary. Considering the presentation and focus of other papers published in this venue, I think this could be seen as common sense. I did not find enough significance, although it may be that no one's done this exact proof before.

Experimental validation: Upright-Net is the only other method compared against. While the numbers are better, I think the experiments are simple enough that a more thorough validation should have been done; using the same backbone network, etc. And I was not sure what the take-home message was; so is it better to predict the upright pose as a matrix and orthogonalize as presented? What was the reason for the improvement? And for the full-orientation prediction, N-way classification is a reasonable approach, but it is not compared against any baseline, and the final pose is dependent on the upright direction prediction which I did not find convincing enough.

**Questions:**

Experiment suggestion: The paper says "our method reliably provides a plausible set of candidate orientations for diverse shapes unseen during training". I think it might be good to do a novel category evaluation experiment.

---

> ### Author Response · Authors · 2024-11-19
> **Rebuttal to Reviewer XS2K (Part 1/2)**
>
> **Part 1/2**
>
> Thank you for your thoughtful review. **Our primary contribution in “Orient Anything” is a shape orientation pipeline that surpasses the previous state of the art by a wide margin on well-established benchmarks in this domain.** In particular, our pipeline outperforms the previous SOTA (Upright-Net) on up-axis estimation by nearly **20 percentage points**, which corresponds to a **64.6% reduction in error rate.** Furthermore, our method is able to reliably return randomly-rotated shapes drawn from all classes in ShapeNet to upright and front-facing orientation; to our knowledge, our method is the first to solve this task. We are able to do this thanks to a novel decomposition of the shape orientation problem into quotient regression and classification, which we support with rigorous theory proving that our method can recover orientations up to symmetries in the input shapes. We contrast this with a naive regression strategy, which fails on rotationally-symmetric shapes; these results shed light on why previous work like Liu et al.’s “Upright orientation of 3D shapes with convolutional networks” have had to resort to strategies such as training $n$ different orienters via regression.
>
> **We have uploaded a revised manuscript**, with new text highlighted in blue. We individually address your concerns below:
>
> *I appreciated the careful explanation and presentation, but spending multiple pages on why a direct L2 regression of rotational symmetry doesn't work seems unnecessary. Considering the presentation and focus of other papers published in this venue, I think this could be seen as common sense. I did not find enough significance, although it may be that no one's done this exact proof before.*
>
> Our results on the failures of L2 regression are not our primary contribution. They contextualize our remaining theoretical results (Propositions 3.3 and 3.4) which show *why* our pipeline is able to succeed where naive approaches have failed. However, we emphasize that our primary contribution is a state of the art shape orientation pipeline that greatly outperforms the previous SOTA **while being backed up by rigorous theory**. As noted by another reviewer, shape orientation is an important problem in 3D shape analysis, and is the subject of a long line of research, with recent work such as Pang et al.’s “Upright-Net” being published in impactful venues such as CVPR. We have further emphasized the role of Propositions 3.1 and 3.2 in our presentation on lines 169-170 of the revised manuscript.
>
> *While the numbers are better, I think the experiments are simple enough that a more thorough validation should have been done; using the same backbone network, etc.*
>
> Our experiments demonstrate that our method outperforms the previous SOTA by a nearly 20-point margin. Responding to another reviewer’s request, we have additionally replicated the shape alignment experiment in Section 4.2 of "Adjoint Rigid Transform Network: Task-conditioned Alignment of 3D Shapes" by Zhou et al. (2022) to the best of our ability in Appendix C.1 of the revised manuscript. 89.1% of our method’s pairwise angular errors are less than 10 degrees, which significantly exceeds the accuracy of ~80% achieved by Zhou et al (2022).
>
> *And I was not sure what the take-home message was; so is it better to predict the upright pose as a matrix and orthogonalize as presented? What was the reason for the improvement?*
>
> Our method succeeds because of our novel two-stage decomposition of the orientation estimation problem into quotient regression and classification. Our theoretical results (Props. 3.1 to 3.4) explain why our method succeeds whereas naive regression approaches fail, and culminate in Prop. 3.4, which shows that our method is able to recover an arbitrarily-rotated shape’s orientation up to one of its symmetries, which is sufficient to recover the canonically-oriented shape.
>
> In contrast to our method, Upright-Net (the previous SOTA) assumes that each shape has a supporting base on which it stands in upright orientation. To orient a shape into an upright position, it solves a segmentation problem to infer this supporting base, fits a plane to the base, and then returns an inward-pointing normal vector to that plane as its estimate of the shape’s up-axis. This necessarily fails for shapes that lack such a supporting base, and we conjecture that this restrictive prior helps explain why Upright-Net underperforms on all of ShapeNet. We have elaborated on this point in lines 427-429 of the revised manuscript.
>
> *And for the full-orientation prediction, N-way classification is a reasonable approach, but it is not compared against any baseline*
>
> Table 1 in Poursaeed et al. (2020) shows that the accuracy of $N$-way classification for orientation estimation decays dramatically with $N$, reaching a value of **1.6%** when one discretizes SO(3) into $N=100$ fixed rotations. We now reference this fact in lines 116-120 of the revised manuscript.

---

> ### Author Response · Authors · 2024-11-19
> **Rebuttal to Reviewer XS2K (Part 2/2)**
>
> **Part 2/2**
>
> *the final pose is dependent on the upright direction prediction which I did not find convincing enough*
>
> We unfortunately do not understand this particular point; could you please elaborate on it? To succeed on full-orientation prediction, our method must by definition correctly predict all three of a shape’s orientation axes (including the upright direction) up to one of its symmetries.
>
> **Conclusion.**
>
> We thank you for your review of our paper and hope that we have addressed your concerns. If you have any remaining questions, we would be pleased to continue this conversation during the discussion phase. Otherwise, **we respectfully ask that you raise your score for this submission.**

---

> ### Comment · Reviewer_XS2K · 2024-11-24
>
> Thank you for your detailed response and for addressing my concerns. I appreciate the effort you've put into revising the manuscript and providing additional clarifications. I'd like to expand on a few points and clarify my position.
>
> First, regarding my comment about the final pose being dependent on the upright direction prediction: I apologize for the confusing wording in my original review. What I meant was that it would have been better to evaluate the performance of the flipper (N-way classification) independently of the orienter. I don't think the dependency itself is unreasonable in the final pipeline. However, from an analysis perspective, such an experiment would provide a clearer understanding of the contribution of each stage in the pipeline. One possible experiment could involve using an oracle model for the upright vector and only evaluating the classifier against some baselines. Why N-way classification? Why not regress sin/cos values or predict a (2d) rotation matrix (similarly to the up-orienter)? Reviewer acsD also asked "how would the Flipper perform without the quotient oriented?"
>
> On the novelty of the approach, I share Reviewer Zyej's concern that hybrid classification+regression strategies are common, and the key observations about symmetries complicating L2 regression are not particularly new. Reviewer acdD also points out  it "could be expressed in a much more direct way." While I don't want to overstate this point, I agree with this sentiment, and these aspects may not stand out as significant contributions. It feels more like an approach that would typically serve as a baseline. My opinion is that theoretical contributions, while present, don't provide enough depth or insight to elevate the paper above the threshold.
>
> Regarding the performance improvement over Upright-Net, while the results are strong, I still question whether this improvement alone is sufficient to justify acceptance. I would prefer it to be more based on what new questions this paper answers and how other researchers can benefit from the findings. The lack of broader experimental validation and analysis, as noted by other reviewers, remains a concern in my opinion.
>
> Finally, the training time of three weeks on an RTX 3090 GPU is surprising given the scale of the approach and dataset described in the paper. This raises questions about whether there are other unexplained factors contributing to this. I don't want to speculate too much, but I currently share this concern with another reviewer who ask similar questions about the training time.
>
> In conclusion, while I appreciate the authors' efforts in revising the manuscript, my overall assessment remains unchanged. Thank you again for your response.

---

### Official Review · Reviewer_Zyej · 2024-11-04

**Soundness:** 3
**Presentation:** 3
**Contribution:** 3
**Rating:** 6
**Confidence:** 3

**Summary:**

This paper proposes a supervised method for predicting the orientation of a 3D object. The main observation is that direct regression will fail due to the abundance of symmetry in 3D shapes. Therefore, a two-stage approach is approached. Using the DGCNN backbone, the method first regresses the orientation up to octahedral symmetries (chosen as it is the single group that contains the most common symmetries in SO3). Then, the method classifies the transfomation that "flips" the shape, oriented by the first step, into the correct orientation. The method is intuitively sound and avoids issues with previous regression based works, by having better overall performance, work on more general settings (full orientation, single model over entire collection of shapes, instead of per catregory), and have error characteristics (centered around few modes) that make further human-in-the-loop correction easier.

**Strengths:**

- Practically useful approach that clearly contributes to an important problem in 3D shape analysis. Clearly outperforms relevant works, achieving better quantitative performance and is more general.
- The idea/intuition behind this approach is not only useful for the particular setting (orientation estimation), but generally applicable to a much wider range of 3D tasks where certain steps involve predictions that can be complicated by symmetries (just naming a few that comes to my mind: part-based analyis/generation, assembly, articulation, shape retrieval, etc.)
- Solid mathematical analysis to back up the method - though I do not have the background to and did not check thoroughly for its correctness. The proofs does sound reasonable.

**Weaknesses:**

- One could argue that the method itself, though wrapped in solid mathematics, is not *that* novel, since hybrid classification+regression strategies are very common overall, and the key observations about symmetries complicating L2 regression is not new. It might help to further discuss/clarify the novelty.
- The octahedral group, while covering a good amount of 3D symmetries, also leave out a lot of important ones e.g. the very common 5-way symmetry for chair bases, many mechnical objects e.g. bolts/gears. While I think there can be extensions to this method to cover an even wider range of shapes, the present version probably would not work well on these. And unlike a single step approach, I do have the feeling that this method will struggle a lot more for these shapes. Some analysis here would be ideal - either on how to sidestep these cases or to show that the proposed method works no worse than single step approaches.
- Arguably, ShapeNet is no longer the go-to dataset now given that there exist much larger shape datasets now. Would be interesting to see more analysis there. Figure 11 does cover objaverse but might be good to have more in the main body.

**Questions:**

I am overall leaning towards acceptance. I think this paper contains a solid, although somewhat straightforward, idea, that can be useful in many applications. I am currently not willing to champion for the paper, as I am personally not too sure about the magnitude of impact this paper can make, and given that there are clearly cases where the proposed method cannot solve (but perhaps can be adapted to solve). More clarification on novely/contribution and more analysis on these potential limitations can make me more positive.

---

> ### Author Response · Authors · 2024-11-19
> **Rebuttal to Reviewer Zyej (Part 1/2)**
>
> **Part 1/2**
>
> Thank you for your thoughtful review. **Our primary contribution in “Orient Anything” is a shape orientation pipeline that surpasses the previous state of the art by a wide margin on well-established benchmarks in this domain.** In particular, our pipeline outperforms the previous SOTA (Upright-Net) on up-axis estimation by nearly **20 percentage points**, which corresponds to a **64.6% reduction in error rate**. Furthermore, our method is able to reliably return randomly-rotated shapes drawn from all classes in ShapeNet to upright and front-facing orientation; to our knowledge, our method is the first to solve this task. We are able to do this thanks to a novel decomposition of the shape orientation problem into quotient regression and classification, which we support with rigorous theory proving that our method can recover orientations up to symmetries in the input shapes. We contrast this with a naive regression strategy, which fails on rotationally-symmetric shapes; these results shed light on why previous work like Liu et al.’s “Upright orientation of 3D shapes with convolutional networks” have had to resort to strategies such as training $n$ different orienters via regression.
>
> We have uploaded a revised manuscript, with new text highlighted in blue. We individually address your concerns below:
>
> *One could argue that the method itself, though wrapped in solid mathematics, is not that novel, since hybrid classification+regression strategies are very common overall, and the key observations about symmetries complicating L2 regression is not new. It might help to further discuss/clarify the novelty.*
>
> The key novel contribution of our paper is a shape orientation pipeline that greatly surpasses the previous state of the art. In contrast, our observations on the impact of symmetries on L2 regression contextualize our theoretical results showing why our pipeline is able to succeed where naive approaches have failed.
>
> *The octahedral group, while covering a good amount of 3D symmetries, also leave out a lot of important ones e.g. the very common 5-way symmetry for chair bases, many mechnical objects e.g. bolts/gears. While I think there can be extensions to this method to cover an even wider range of shapes, the present version probably would not work well on these. And unlike a single step approach, I do have the feeling that this method will struggle a lot more for these shapes. Some analysis here would be ideal - either on how to sidestep these cases or to show that the proposed method works no worse than single step approaches.*
>
> Our method can easily be adapted to any symmetry group. Because our theory is general, our theorems hold for any subgroup $\hat{\mathcal{R}}$ of $SO(3)$ containing the symmetries of the input shapes. In practice, implementing our method with a different choice of $\hat{\mathcal{R}}$ is as simple as generating a set of rotation matrices representing the symmetries in $\hat{\mathcal{R}}$ and retraining our model. This can be done in a few lines of code. Exposing this design choice in a user-friendly manner is among our paper’s novel contributions and enables our method to be adapted to specialized datasets. We have elaborated on this point in Section 3.2 of the revised manuscript (see lines 328-331).
>
> Furthermore, existing shape orientation methods impose *more restrictive* priors on the input shapes than ours. For example, Upright-Net (the previous SOTA) assumes that each shape has a supporting base on which it stands in upright orientation. To orient a shape into an upright position, it solves a segmentation problem to infer this supporting base, fits a plane to the base, and then returns an inward-pointing normal vector to that plane as its estimate of the shape’s up-axis. This necessarily fails for shapes that lack such a supporting base, and we conjecture that this restrictive prior helps explain why Upright-Net underperforms on all of ShapeNet. We make this explicit in Section 4.1 of the revised manuscript (see lines 427-429).
>
> **Continued in Part 2/2 below.**

---

> ### Author Response · Authors · 2024-11-19
> **Rebuttal to Reviewer Zyej (Part 2/2)**
>
> **Part 2/2**
>
> *Arguably, ShapeNet is no longer the go-to dataset now given that there exist much larger shape datasets now. Would be interesting to see more analysis there. Figure 11 does cover objaverse but might be good to have more in the main body.*
>
> We benchmarked our method’s performance on ShapeNet because it is the largest and most diverse dataset we are aware of whose shapes are canonically oriented. We cannot quantitatively benchmark our method on Objaverse because its shapes are in arbitrary orientations; we consequently lack a consistent ground truth orientation for each shape against which to compare our method’s inferred orientations. Given the current level of interest in 3D deep learning, we anticipate that larger datasets of canonically oriented shapes will soon be released, giving us the opportunity to train and benchmark our pipeline on larger datasets. We now further emphasize this point in the introduction to Section 4 in the revised manuscript (see lines 400-402).
>
> *I am personally not too sure about the magnitude of impact this paper can make*
>
> As you remark, shape orientation is “an important problem in 3D shape analysis” and has been the subject of a long line of research, with recent work such as Pang et al.’s “Upright-Net” being published in impactful venues such as CVPR. Furthermore, as you have noted, our work “clearly outperforms relevant works, achieving better quantitative performance and being more general.” You further observe that our strategy is “generally applicable to a much wider range of 3D tasks where certain steps involve predictions that can be complicated by symmetries.” We appreciate your positive assessment of our contributions and believe that it indicates this paper will have a substantial impact on 3D shape analysis. As we intend to open-source our code and model weights, we also anticipate that our method will serve as a useful tool for the community.
>
> **Conclusion.**
>
> We thank you for your careful review and hope that we have addressed your concerns. If you have any remaining questions, we would be pleased to continue this conversation during the discussion phase. Otherwise, **we respectfully ask that you raise your score for this submission.**

---

> > ### Comment · Reviewer_Zyej · 2024-11-25
> >
> > A couple quick questions:
> >
> > 1. "Our method can easily be adapted to any symmetry group. Because our theory is general, our theorems hold for any subgroup ...": What about cases where a collection of shapes need to be represented multiple subgroups? It is non trivial to just select one, even with a user interface, since one needs to figure out the shape <-> subgroup correspondance per shape, and then train multiple models for each of the group. How is this practical? I guess one can incorporate an additional module that predicts the subgroup per shape, or directly detects symmetry and choose the subgroup accordingly. However, neither is that trivial, and I would like to see some discussion/evidence here.
> >
> > 2. "This necessarily fails for shapes that lack such a supporting base, and we conjecture that this restrictive prior helps explain why Upright-Net underperforms on all of ShapeNet. ". A few qualitative examples will help greatly with this claim.

---

### Author Response · Authors · 2024-11-27
**Joint response to reviewers**

We are grateful for the reviewers’ valuable comments and their continued engagement throughout the discussion period. We have uploaded a final revision of the manuscript, where we have fixed the caption for Figure 9. (We thank Reviewer acsD for pointing out this issue.) Below, we outline our final case for this work’s publication.

The reviewer guidelines state that “different objectives [...] require different considerations as to potential value and impact.” Our objective in “Orient Anything” is to build a shape orientation pipeline that surpasses the previous state of the art by a wide margin on well-established benchmarks in this domain. **All reviewers agree that we have delivered on our objective**, with Reviewer Zyej stating that Orient Anything “clearly outperforms relevant works,” Reviewer XS2K stating that our results are “strong,” and Reviewer acsD stating that we achieve “significantly better accuracy on ShapeNet and ModelNet.” We emphasize that our performance improvements are far from marginal: We exceed the previous SOTA (Upright-Net) on up-axis estimation by nearly 20 percentage points, corresponding to a 64.6% reduction in the error rate. As noted by Reviewer Zyej, our method is also “more general” than competing methods, which we demonstrate by training and testing our pipeline on all of Shapenet, rather than a subset of classes as in previous work.

Reviewers Zyej and acsD agree on the importance of the shape orientation problem, with the former characterizing it as an “important problem in 3D shape analysis,” and the latter calling it a “valuable problem that could potentially impact many different research areas.”  Reviewer XS2K asks “how other researchers can benefit from [our] findings.” Zyej believes that our two-stage strategy is “generally applicable to a much wider range of 3D tasks where certain steps involve predictions that can be complicated by symmetries,” and cites tasks such as part-based analysis and generation, assembly, articulation, and shape retrieval as examples. In light of its strong performance and reproducibility, acsD also anticipates that Orient Anything “could be an instrumental tool for several downstream tasks.” We believe this indicates that the shape analysis and 3D deep learning communities will benefit greatly from our contributions.

Some reviewers shared concerns about our method’s simplicity, with Zyej stating that “hybrid classification+regression strategies are very common overall” and XS2K stating that our method “feels more like an approach that would typically serve as a baseline.” Our method’s simplicity is among its *key advantages*, making theory tractable, which enables us to back up our method’s SOTA performance by rigorous results that help explain its empirical success. To our knowledge, there is no prior work in shape orientation employing our two-stage problem decomposition; as such, Orient Anything is a clearly novel contribution to this literature.

The reviewers’ comments have shown that Orient Anything is a **novel** algorithm for solving an **important** problem in 3D shape analysis, **achieving SOTA by a wide margin** while being **more general** than previous approaches. We believe this suggests that our work will have a substantial impact on shape analysis and 3D deep learning, serve as a useful tool to the community, and form a valuable part of this year’s ICLR program.

---

### Meta-Review · Area_Chair_XvjV · 2024-12-21

**Metareview:**

**Summary**

The paper proposes to predict the canonical orientation of a 3D object represented as a point cloud.  A two stage pipeline is proposed: 1) an orienter (DGCNN) regressor network that identifes the orientation (up to octahedral symmetries)  2) a flipper (classification network) that selects among the symmetries (24 for octahedral group).  Experiments on up prediction, and full-orientation prediction are conducted to show the effectiveness of the proposed approach.

The main contributions of the work are: 1) revisiting the task of canonical pose estimation using modern techniques 2) proposal of a two stage approach that takes into account that many shapes contains rotational symmetries, 3) mathematical analysis of the soundness of the proposed method.

**Strengths**

1. The paper addresses a practical problem of orienting 3D objects. [acsD]
2. Proposed approach is shown to perform prior work [Zyej]
3. Solid mathematical analysis backing up the method [Zyej,acsD]
4. Idea is interesting and can potentially be useful for other 3D tasks involving dealing with shape symmetries [Zyej,acsD]
5. Code is provided [acsD]

**Weaknesses**

1. Lack of discussion on what constitutes "canonical orientation" across different object categories. As noted by reviewer acsD, there are ambiguities on what would be considered the "canonical orientation".
   - For instance, for a "book", is the "canonical orientation" the book with the front cover of the book as the front (as typically placed on desks) or the spine of the book as the front (as typically found on bookcases).  For a L-shaped couch, which side would be considered the front?
2. Weak experiments and lack of ablation and analysis [XS2K,acsD]
   - No comparisons / baselines for full-orientation estimation.  For instance, a class-specific method can be used as a comparison point.
3. Presentation needs improvement [XS2K,acsD]
   - Overcomplicated long explanation in the main paper of why L2 loss is not ideal [XS2K]
   - Qualitative visualizations are poorly quality and hard to read [acsD]
4. Training cost and how well would the model scale [acsD]
5. Limited discussion on various design decisions (octahedral group) [acsD]

**Recommendation**

The AC believe predicting the canonical pose of a 3D object can be useful in many settings.  The paper presents an interesting approach that can potentially be applied to another shape-analysis problems that involves dealing with shape symmetries.

Overall, while the work has strong mathematical foundations, reviewers felt there was limited discussion on the problem itself (e.g. whether it makes sense for a canonical orientation across object categories) and limited experiments. Reviewers also expressed the opinion that the presentation can be improved so that the paper is easier to read.  Finally, the AC note that while the problem statement is important for different applications, the topic may be more appropriate for a graphics or vision venue.

Based on the above and reviewer ratings, the AC feels that the work is not ready for acceptance at ICLR.  The AC encourages the authors to improve the work and consider resubmitting to a future venue.

**Additional notes**

A bit of discussion to address weakness 1, and some improved experiments (including perhaps metrics that take into account that some orientations are equally valid), together with improved presentation will make this work much more compelling.
For instance, the problem statement and proposed solution can be stately in a more concise and simple manner, with some details moved to the appendix.  It would also be good to discuss limitations of the work.

In addition to the above, the AC notes the following (minor points):
- The paper refers to solutions to different problems (e.g. "Problem 2").  The AC believe these refer to the numbered equations in the paper and recommends referring to these as "Equation 2", etc.
- The notation for L2 regression uses "L^2", where it is more common to use "L_2"

**Additional Comments On Reviewer Discussion:**

The paper initially received two rejects (one 3 and one 5) and one marginal accept (score of 6).  After the author response, one reviewer (XS2K) increased their score to 5 (from 3).

While some reviewer concerns were addressed by the rebuttal, reviewers were still lukewarm / borderline negative on the work.  In particular, concerns about how the canonical orientation is defined across object categories, presentation quality, and limited experiments were not fully addressed.  Reviewers also expressed concerns about the training time of the method.

Due to these concerns, reviewers felt the work was not ready for presentation at ICLR.

---

### Decision · Program_Chairs · 2025-01-22

Reject